# The Iterative Optimal Brain Surgeon: Faster Sparse Recovery by Leveraging Second-Order Information

**Diyuan Wu**[1] **Ionut-Vlad Modoranu**[1] **Mher Safaryan**[1]

**Denis Kuznedelev**[2,3] **Dan Alistarh**[1]

[1]Institute of Science and Technology Austria (ISTA) [2]Yandex Research [3]Skoltech

## Abstract

The rising footprint of machine learning has led to a focus on imposing *model sparsity* as a means of reducing computational and memory costs. For deep neural networks (DNNs), the state-of-the-art accuracy-vs-sparsity is achieved by heuristics inspired by the classical Optimal Brain Surgeon (OBS) framework [LeCun et al., 1989, Hassibi and Stork, 1992, Hassibi et al., 1993], which leverages loss curvature information to make better pruning decisions. Yet, these results still lack a solid theoretical understanding, and it is unclear whether they can be improved by leveraging connections to the wealth of work on sparse recovery algorithms. In this paper, we draw new connections between these two areas and present new sparse recovery algorithms inspired by the OBS framework that comes with theoretical guarantees under reasonable assumptions and have strong practical performance. Specifically, our work starts from the observation that we can leverage curvature information in OBS-like fashion upon the projection step of classic iterative sparse recovery algorithms such as IHT. We show for the first time that this leads both to improved convergence bounds under standard assumptions. Furthermore, we present extensions of this approach to the practical task of obtaining accurate sparse DNNs, and validate it experimentally at scale for Transformer-based models on vision and language tasks.

## 1 Introduction

The increased focus on efficiency in machine learning has led to significant interest in *sparsity* as a means of reducing both the computational and memory costs of training and executing accurate models, with literally hundreds of methods being proposed recently [Hoefler et al., 2021].The key trade-off in this context is between the degree of sparsity imposed and the accuracy of the dense (uncompressed) model.

Currently, the best performing *post-training* methods for DNNs are inspired by the Optimal Brain Damage (OBD) framework of LeCun et al. [1989], generalized via the Optimal Brain Surgeon (OBS) approach of Hassibi and Stork [1992], Hassibi et al. [1993].[1] In brief, OBD/OBS poses accurate pruning as a constrained optimization problem, and approximates the optimal solution by leveraging second-order information about the loss function to make better pruning decisions. Recently, there has been significant work on versions of OBS that can scale to deep neural networks (DNNs), leading to surprisingly good results at scale [Frantar and Alistarh, 2022, 2023]. Yet, no theoretical guarantees are known for OBS-type algorithms.

---

[1]The two frameworks primarily differ in the second-order approximation. Since OBS is a strict generalization of OBD, in the following we use OBS to denote the pruning framework.

By contrast, the area of *sparse recovery* provides a rich set of theoretically-justified iterative algorithms for reconstructing or approximating a high-dimensional but sparse signal from a limited set of observations, such as Iterative Hard Thresholding (IHT) [Blumensath and Davies, 2009] or Matching Pursuit (MP)-type [Tropp and Gilbert, 2007] algorithms. Yet, to our knowledge, these algorithms do not usually employ curvature information in their structure, and do not extend to DNNs.

It is still an open question whether one can connect these lines of work, for instance by leveraging second-order information to improve the convergence of sparse recovery algorithms, or by extending these algorithms to the more complex task of sparsifying multi-layer DNNs. This is the question we approach in this paper.

**Contributions.** In this paper, we propose a family of algorithms that generalize the post-training OBS framework to the iterative setting, popular in the context of sparse recovery. This new approach, which we call the Iterative Optimal Brain Surgeon (I-OBS), provides convergence guarantees that can improve upon classic algorithms such as IHT by leveraging second-order information on the sparse projection step. As a consequence, we provide the first theoretical guarantees for popular practical pruning heuristics such as OBC Frantar and Alistarh [2022] or WoodFisher [Singh and Alistarh, 2020]. Second, on the practical side, we show that the improved guarantees can lead to faster practical convergence.

In more detail, our contributions are as follows:

- We propose a new family of sparse recovery algorithms called Iterative Optimal Brain Surgeon (I-OBS), which generalize upon the classic IHT-based algorithms by leveraging approximate second-order information in the sparse projection step.

- We prove that this family of algorithms provides faster analytical convergence rates than previously-known methods, under standard first- and second-order smoothness and strong convexity assumptions. We also show that prior practical pruning algorithms such as WoodFisher [Singh and Alistarh, 2020] and OBC [Frantar and Alistarh, 2022] are special cases of our framework, providing them with theoretical guarantees.

- We provide practical variants of these algorithms, which relax the theoretical constraints but can be easily implemented and can scale to large problem instances arising in the compression of vision and language models. We validate our results both on synthetic datasets, showing that our approach leads to faster convergence relative to IHT, and therefore validating our theoretical findings. Finally, we present empirical evidence that our approach can scale to large models and lead to improved results, by implementing it to accurately sparsify models with more than 1 billion parameters.

## 2 Related Work

**Model compression.** In recent years, the literature has witnessed a significant surge in research focusing on model compression. Among various techniques, network pruning or sparsification stands out as a powerful approach, effectively reducing the model's parameter count by eliminating the least influential weights and fine-tuning the remaining ones [Singh and Alistarh, 2020, Lin et al., 2020, Hoefler et al., 2021, Frantar and Alistarh, 2023]. Following a similar concept to pruning, quantization techniques suggest reducing the precision of each parameter by allocating fewer bits, rather than completely removing weights from the network [Yao et al., 2022, Dettmers et al., 2022, Frantar et al., 2023, Kim et al., 2023]. Additionally, methods that leverage low-rank factorization are gaining popularity due to their inherent tensor structure [Hu et al., 2022, Lialin et al., 2024, Xu et al., 2023, Nikdan et al., 2024]. Notably, these compression methods are versatile, applicable during both the optimization and post-training phases. Another noteworthy technique is knowledge distillation [Hinton et al., 2014, Mishra and Marr, 2018, Polino et al., 2018], where the compressed model is a compact "student" model trained by a larger "teacher" model. The teacher model can either be already trained to high accuracy or trained simultaneously with the student model [Guo et al., 2020, Harutyunyan et al., 2023].

**Sparse recovery.** The model compression problem we study in this paper is closely related to the field of sparse recovery, which aims to optimize certain objective function based on sparsity constraints, see e.g. [Zhao, 2018] for a detailed introduction. One of the most common sparse optimization algorithms is the iterative hard thresholding algorithm (IHT) first proposed by Blumensath and Davies

[2009] for compressed sensing problem, which applies a top$k$ operator to each gradient update, which keeps the $k$ elements with the largest magnitude and setting the rest to zero. While IHT is simple and effective, there is also a long line of work [Blumensath, 2012, Bahmani et al., 2013, Chen and Gu, 2017, Meng and Zhao, 2020, Zhou et al., 2021] that studies the sparse variants of Newton's methods for these sparse optimization problems, aiming to get acceleration benefits. In particular, Blumensath [2012] proposed a general definition of accelerated IHT for sparse linear regression problems, and showed an accelerated convergence rate in this type of method. Bahmani et al. [2013] proposed a general framework called GraSP, and they proposed to use a restricted Newton step as an approximation of the sparse constrained optimization problem. Chen and Gu [2017] proposed a Fast Newton Hard Thresholding Pursuit, which uses an iterative algorithm to estimate the inverse Hessian matrix for a stochastic objective function. They also obtain a composite rate of convergence. Meng and Zhao [2020] consider applying hard thresholding to Newton's method for sparse linear regression problems, where they call Newton-Step-Based Iterative Hard Thresholding (NSIHT), and prove a global convergence for this specific problem. Zhou et al. [2021] proposed a general framework called Newton Hard Thresholding Pursuit (NHTP) for general sparse optimization problems showing implicit global and local quadratic convergence rates. In contrast to our approach, at each step NHTP selects the spare support based on the first-order gradient information, switches to restricted gradient update whenever the restricted Newton update fails to provide sufficiently good descent direction, and additionally incorporates a line search procedure to achieve global convergence.

**Further comparison.** Our work follows the salience-based weight pruning approach, which evaluates the impact of removing weights on the model's loss or output. Among these, methods based on second-order information are most relevant, particularly those following the Hassibi et al. [1993] framework. Dong et al. [2017] use a second-order approximation of the loss function, assuming a zero gradient, and introduce a layerwise strategy to approximate the Hessian. Singh and Alistarh [2020] scale this idea with the empirical Fisher approximation of the Hessian, improving performance and considering non-zero gradients. However, WoodTaylor methods by Singh and Alistarh [2020] focus on pruning one weight at a time, whereas our work extends this to multiple weights.

Other methods addressing multiple weight pruning include Combinatorial Brain Surgeon (CBS) [Yu et al., 2022], which formulates mask selection as an integer programming problem and proposes two heuristics to approximate its solution. Frantar and Alistarh [2022] tackle layerwise pruning with a quadratic problem and introduces a greedy heuristic for efficient layerwise problem-solving. Similarly, Benbaki et al. [2023] formulate layerwise pruning as a $\ell_0\ell_2$-regularized quadratic problem and propose an IHT-like iterative algorithm with a line search strategy. Aghasi et al. [2017] minimize the norm of weight matrices while ensuring similar activations in the pruned layer. While prior iterative algorithms like CHITA [Benbaki et al., 2023] focus on specific problems, our methods apply to general loss functions beyond quadratic problems and offer convergence guarantees. Practically, within our iterative framework, using a cost-effective projection step (e.g., TopK or SparseGPT [Frantar and Alistarh, 2023]) yields competitive results compared to more complex one-shot solvers.

## 3 Derivation and convergence of I-OBS

**3.1. Notation.** Denote $[d] = \{1, 2, \ldots, d\}$. For a given vector $\theta \in \mathbb{R}^d$, let $\mathrm{supp}(\theta) = \{i \in [d] : \theta[i] \neq 0\}$ be the support of $\theta$, and $\|\theta\|_0 = |\mathrm{supp}(\theta)|$ be the cardinality of its support, where $|\cdot|$ denotes the cardinality of a set. Let $\{e_1, e_2, \ldots, e_d\}$ be the standard basis vectors in $\mathbb{R}^d$. For a given index set $Q \subseteq [d]$, we define $e_Q = \sum_{q \in Q} e_q$, and $(\theta)_Q = \sum_{q \in Q} \theta[q] e_q$ the restriction of $\theta$ to $Q$. We denote $\theta \odot \theta'$ to be the Hadamard product of vectors $\theta$ and $\theta'$. Finally, we denote $T_k$ the "Top-$k$" operator choosing the top $k$ coordinates of the input by their magnitude and zeroing out the rest.

Next, we introduce notation for manipulating matrices by removing specific rows and columns. Given an index set $Q \subseteq [d]$, we define the square matrix $\mathbf{E}_Q \in \mathbb{R}^{d \times d}$ with diagonal entries $(\mathbf{E}_Q)_{i,i} = 1$ for $i \in Q$, and all other entries set to zero. Consequently, for a matrix $\mathbf{H} \in \mathbb{R}^{d \times d}$, the result of the left multiplication $\mathbf{E}_Q \mathbf{H} \in \mathbb{R}^{d \times d}$ yields a matrix where only the rows with indices in $Q$ are retained, and all other rows are set to zero. Similarly, the right multiplication $\mathbf{H} \mathbf{E}_Q \in \mathbb{R}^{d \times d}$ keeps the columns with indices in $Q$ untouched, and sets all other columns to zero.

Analogously, for a given index set $Q = \{q_1, ..., q_{|Q|}\} \subseteq [d]$, define the rectangular matrix $\mathbf{I}_Q \in \mathbb{R}^{|Q| \times d}$ as $\mathbf{I}_Q = [e_{q_1}^\top \ e_{q_2}^\top \ \ldots \ e_{q_{|Q|}}^\top]^\top$. Hence, for a matrix $\mathbf{H} \in \mathbb{R}^{d \times d}$, the resulting matrix

$\mathbf{I}_Q \mathbf{H} \in \mathbb{R}^{|Q| \times d}$ is composed of the rows with indices in $Q$. Similarly, $\mathbf{H}\mathbf{I}_Q^\top \in \mathbb{R}^{d \times |Q|}$ is composed of columns with indices in $Q$. Finally, given a real symmetric matrix $\mathbf{H} \in \mathbb{R}^{d \times d}$, we denote the eigenvalues of $\mathbf{H}$ as $\lambda_1(\mathbf{H}) \leq \ldots \leq \lambda_d(\mathbf{H})$, dropping $\mathbf{H}$ in the notation when clear from context.

**3.2. Problem setup and assumptions.** We consider a general optimization problem with sparsity constraints:

$$\min_{\theta \in \mathbb{R}^d} f(\theta) \quad \text{subject to} \quad \|\theta\|_0 \leq k, \tag{1}$$

where $f : \mathbb{R}^d \to \mathbb{R}$ is a given smooth objective function of parameters $\theta \in \mathbb{R}^d$ and $k$ is the sparsity threshold. Finding or approximating the solution of (1) is known to be NP-hard even for quadratics [Natarajan, 1995, Foster et al., 2015].

We make use of the following assumptions regarding the convexity and smoothness of the objective function, which are similar to those in e.g. [Peste et al., 2021]:

**Assumption 1** (The existence of a sparse solution)**.** The function $f(\theta)$ admits a minimizer $\theta^*$ with sparsity $k^* \leq d$.

**Assumption 2** ($\mu$-Strong convexity)**.** The function $f \colon \mathbb{R}^d \to \mathbb{R}$ is differentiable and $\mu$-strongly convex for some constant $\mu > 0$, i.e., for any $\theta, \theta' \in \mathbb{R}^d$ it holds

$$f(\theta) \geq f(\theta') + \langle \nabla f(\theta'), \theta - \theta' \rangle + \frac{\mu}{2}\|\theta - \theta'\|_2^2. \tag{2}$$

**Assumption 3** ($(k, d-k, L)$-Restricted first-order smoothness)**.** The function $f : \mathbb{R}^d \to \mathbb{R}$ is twice differentiable and $k$-restricted $L$-smooth for some constant $L > 0$, i.e., for any $\theta \in \mathbb{R}^d$ such that $\|\theta\|_0 \leq k$ it holds that:

$$v^\top \nabla^2 f(\theta) v \leq L\|v\|_2^2, \quad \text{for all } v \text{ with sparsity } \|v\|_0 \leq d - k. \tag{3}$$

As our approach relies on curvature information, it is common to assume Hessian smoothness, similar to the gradient smoothness Assumption 3 in first-order optimization.

**Assumption 4** ($(2k + k^*, M)$-Restricted second-order smoothness)**.** The function $f \colon \mathbb{R}^d \to \mathbb{R}$ is twice differentiable and $(2k + k^*)$-restricted $M$-smooth for some constant $M > 0$, i.e., for any $\theta, \theta' \in \mathbb{R}^d$ such that $\|\theta\|_0 + \|\theta'\|_0 \leq 2k + k^*$ it holds

$$\|\nabla^2 f(\theta) - \nabla^2 f(\theta')\|_2 \leq M\|\theta - \theta'\|_2, \tag{4}$$

where the norm for a matrix $A$ is defined as $\|A\|_2 = \max_{\|x\|_2=1} \|Ax\|_2 = \sqrt{\lambda_{\max}(A^\top A)}$.

When the function is twice differentiable, note that the strong convexity (2) is equivalent to $\|\nabla^2 f(\theta)\|_2 \geq \mu$ for all $\theta \in \mathbb{R}^d$. Also, the $(k, d-k, L)$-Restricted first-order smoothness in Assumption 3 is weaker than $L$-smoothness given the function is twice differentiable, since $L$-smoothness require $v^\top \nabla^2 f(\theta) v \leq L\|v\|_2^2$, for all $v$. Similarly, the $(2k + k^*, M)$-Restricted second-order smoothness assumption is weaker than the usual $\mu$-second-order smoothness which requires $\|\nabla^2 f(\theta) - \nabla^2 f(\theta')\|_2 \leq M\|\theta - \theta'\|_2, \forall \theta, \theta' \in \mathbb{R}^d$

**3.3. IHT as a Proximal Point Method.** Proximal Point Methods (PPMs) are the backbone of many optimization procedures for solving problem (1). In each step, given the current parameters $\theta_t$, PPM defines the next iterate via

$$\theta_{t+1} = \arg\min_{\theta \in \mathbb{R}^d} f(\theta) + \frac{1}{2\eta}\|\theta - \theta_t\|^2, \tag{5}$$

for some learning rate $\eta > 0$. The intuition behind adding a quadratic penalty term is to ensure that the next update $\theta_{t+1}$ is not far away from the current $\theta_t$. Clearly, we can consider PPM with sparsity constraint as in (1). However, solving (5) with or without a sparsity constraint is no easier than the initial problem of minimizing $f(\theta)$.

To make this practical, we adopt a model-based viewpoint, where in each iteration we choose a (possibly stochastic) model $\phi_t(\theta)$ that approximates the objective function $f(\theta)$ in expectation, namely $\mathbb{E}[\phi_t(\theta)] \approx f(\theta)$. For example, we might choose our model $\phi_t(\theta) = f(\theta_t) + \langle \nabla f(\theta_t), \theta - \theta_t \rangle$ to be the linear part of the objective; together with the sparsity constraints, we arrive at

$$\theta_{t+1} = \arg\min_{\theta : \|\theta\|_0 \leq k} \phi_t(\theta) + \frac{1}{2\eta}\|\theta - \theta_t\|^2. \tag{6}$$

With this linear viewpoint, the problem (6) can be solved in closed form and we get the well-known *Iterative Hard Thresholding (IHT)* method [Blumensath and Davies, 2009].

**Lemma 1.** *The closed-form solution to (6) with linear model $\phi_t(\theta) = f(\theta_t) + \langle \nabla f(\theta_t), \theta - \theta_t \rangle$ is*

$$\theta_{t+1} = T_k(\theta_t - \eta \nabla f(\theta_t)).$$

The derivation of $k$-IHT is standard, and we provide the proof of Lemma 1 in Appendix B.2 for completeness. Note that, choosing the stochastic linear model $\phi_t(\theta) = f(\theta_t) + \langle g(\theta_t), \theta - \theta_t \rangle$ with stochastic gradients $g(\theta_t)$, the update (6) reduces to stochastic IHT [Peste et al., 2021]. Besides, dropping the sparsity constraints recovers the standard stochastic gradient descent (SGD).

**3.4. I-OBS: Leveraging Second-order Information.** Our intuition is that we can go beyond the standard Euclidean norm in (6) used in the regularizer, penalizing the distance from the current iterate, by incorporating local curvature information via the Hessian matrix $\mathbf{H}_t = \nabla^2 f(\theta_t)$ instead of the implicit scaled identity matrix $\frac{1}{\eta}\mathbf{I}$, which treats all coordinates equally and ignores correlations. Specifically, we can write:

$$\theta_{t+1} = \arg\min_{\theta:\|\theta\|_0 \leq k} \phi_t(\theta) + \tfrac{1}{2}\|\theta - \theta_t\|_{\mathbf{H}_t}^2. \tag{7}$$

Thus, in each iteration, we minimize the second-order approximation of $f(\theta)$ at $\theta_t$ over the sparse non-convex domain. In turn, this leads us to the proposed Iterative OBS (I-OBS) method. Dropping the constant term $f(\theta_t)$ of $\phi_t(\theta)$, I-OBS can be equivalently defined by

$$\theta_{t+1} = \arg\min_{\theta:\|\theta\|_0 \leq k} \langle \nabla f(\theta_t), \theta - \theta_t \rangle + \tfrac{1}{2}\|\theta - \theta_t\|_{\mathbf{H}_t}^2. \tag{8}$$

We now introduce our idealized algorithm, which we call the Iterative Optimal Brain Surgeon (I-OBS), corresponding to Algorithm 1 with Option 2. The main intuition behind the algorithm is that the computation of $\theta_{t+1}$ in Eqn. (8) can be split into two phases: finding the optimal mask and then solving the problem with a given mask. We present the pseudocode below:

---

**Algorithm 1** Iterative Optimal Brain Surgeon (I-OBS)

1: **Input:** initial value $\theta_1 \in \mathbb{R}^d$, sparsity threshold $k \in [d]$
2: **for** each step $t \in \{1, 2, \ldots, T\}$ **do**
3:      Compute the gradient $g_t = \nabla_\theta f(\theta_t)$ and the Hessian $\mathbf{H}_t = \nabla_\theta^2 f(\theta_t)$,
4:      Compute dense Newton's update $\theta_t^+ = \theta_t - \mathbf{H}_t^{-1} g_t$
5:      ▷ Select the support/mask $Q_{t+1}$ of size $k$ or $d - k$ indices $S_{t+1}$ to prune for the next iterate
6:      *Option 1 (practical):* $Q_{t+1} = \operatorname{supp} T_k(\theta_t^+)$
7:      *Option 2 (theoretical):* with $\mathbf{H}_t^S := \mathbf{I}_S^\top \left( \mathbf{I}_S \mathbf{H}_t^{-1} \mathbf{I}_S^\top \right)^{-1} \mathbf{I}_S$, set $Q_{t+1} = [d] \setminus S_{t+1}$ where

$$S_{t+1} = \arg\min_{S:|S|=d-k} (\theta_t^+)^\top \mathbf{H}_t^S (\theta_t^+),$$

8:      ▷ Optimize the parameters over the selected support/mask $Q_{t+1}$
9:      *Option 1 (practical):* $\theta_{t+1} = (\theta_t^+)_{Q_{t+1}} = T_k(\theta_t^+)$
10:      *Option 2 (theoretical):* $\theta_{t+1} = \left( \mathbf{I} - \mathbf{H}_t^{-1} \mathbf{H}_t^{S_{t+1}} \right) \theta_t^+$
11: **end for**

---

At each step of the algorithm, we first compute Newton's update $\theta_t^+ = \theta_t - \mathbf{H}_t^{-1} g_t$, which might be dense in general. Next, we proceed to select the support $Q_{t+1}$, or equivalently, the set of indices $S_{t+1}$ for pruning. Unfortunately, the selection of the theoretically optimal support set $Q_{t+1}$ as outlined in *Option 2* is intractable due to sparsity constraints. To address this limitation, we introduce the practical *Option 1*, which prioritizes parameters with the largest magnitudes.

Once the mask $Q_{t+1}$ is determined, we optimize the remaining parameters to minimize loss. For *Option 1*, no further optimization is undertaken, and the subsequent iterate $\theta_{t+1}$ comprises just the top $k$ elements with the largest absolute value of Newton's update $\theta_t^+$. On the other hand, the theoretical *Option 2* adjusts the unpruned part of $\theta_t^+$ to precisely solve the sub-problem (8), as demonstrated in the lemma below.

**Lemma 2.** *If $\mathbf{H}_t$ is positive definite, then each step of Algorithm 1 (Option 2) solves Eqn.(8).*

The iterative procedure Algorithm 1 with Option 2, is fairly complex to implement in practice for large models. However, it has a number of interesting theoretical properties and inspires practical extensions, which we describe in the next section. For the procedure Algorithm 1 with Option 1, we call it Top$k$-I-OBS, which we will discuss in more detail in section 3

Although it appears quite theoretical, the I-OBS algorithm is closely related to popular neural network pruning algorithms such as WoodFisher, WoodTaylor [Singh and Alistarh, 2020] and OBC [Frantar and Alistarh, 2022]. In particular, both WoodFisher/WoodTaylor and OBC are special cases of the I-OBS algorithm.

**Connection to WoodFisher/WoodTaylor [Singh and Alistarh, 2020]**   To recover the WoodFisher, we set $\nabla f(\theta_0) = 0$ and $k = d - 1$, obtaining [Singh and Alistarh, 2020, equation (10) and (11)]. More specifically, in this case we have $\theta_0^+ = \theta_0$ and $S = \{i\}$ is a singleton. Hence, the matrix $\mathbf{H}_0^S$ can be computed as $\mathbf{H}_0^S = \frac{e_i e_i^\top}{(H_0^{-1})_{ii}}$. Thus, the criterion of choosing the optimal index to prune becomes $i = \arg\min_{i \in [d]} \frac{\theta_0^2[i]}{(H_0^{-1})_{ii}}$, which recovers [Singh and Alistarh, 2020, equation (11)]. Also, the update of the remaining coordinates are computed as $\theta_1 - \theta_0 = \mathbf{H}_0^{-1} \frac{e_i e_i^\top}{(H_0^{-1})_{ii}} \theta_0 = \frac{-\theta_0[i] \mathbf{H}_0^{-1} e_i}{(H_0^{-1})_{ii}}$, which recovers [Singh and Alistarh, 2020, equation (10)]. This implies that WoodFisher can be considered as running I-OBS for one step under the assumption that $\nabla f(\theta_0) = 0$, and pruning one coordinate. Without the zero gradient assumption, we obtain the WoodTaylor method [Singh and Alistarh, 2020, Eqn. (23)]. The computation is almost the same as the one for recovering WoodFisher, so we omit the details here.

**Connection to OBC [Frantar and Alistarh, 2022]**   To recover the OBC, consider the objective function $f(\theta) = \|\mathbf{X}\theta - \mathbf{X}\theta_0\|_2^2$, where $\mathbf{X} \in \mathbb{R}^{n \times d}$ and $\theta, \theta_0 \in \mathbb{R}^d$. From a practical point of view, we consider $\mathbf{X}$ as the input of a linear layer, and $\theta$ as a row-vector of the weight matrix. Note that a special property of this problem is that the objective function is quadratic, which implies that the Algorithm 1 converges in one step. Indeed, since the objective is quadratic, for any $\theta$ and $\theta_0$, we have $f(\theta) = f(\theta_0) + \langle \nabla f(\theta_0), \theta - \theta_0 \rangle + \frac{1}{2} \|\theta - \theta_0\|_2^2$. However, the difficulty is that we need to find the optimal mask, which might be infeasible in practice. The OBC method [Frantar and Alistarh, 2022, Algorithm 1] applies a greedy strategy, wherein it sets $k = d - 1$ in the first step. Applying the I-OBS algorithm 1 and brute-forcing over all the masks, we find the optimal coordinate to prune. Since, in this case, we only prune one entry at a time, brute-forcing over all the masks takes $\mathcal{O}(d)$ steps, which is feasible. Then, one applies this procedure for $d - k$ times to prune the rest of the weights. Thus, the OBC method can be viewed as running I-OBS for $d - k$ rounds, and for each round one lets the sparsity be $d - 1$, i.e., only pruning one entry at a time.

**3.5. Local convergence of I-OBS.** In this section, we present the local convergence of the I-OBS method described in Section 3 under the regularity Assumptions 1–4. As described in Algorithm 1, it contains a gradient step preconditioned by the Hessian matrix. Thus, our I-OBS method can be viewed as a sparse variant of Newton's method. Below, we demonstrate that, perhaps surprisingly, the imposed sparsity constraint does not degrade the asymptotic local quadratic convergence rate of Newton's method.

**Theorem 1.** *Let Assumptions 1, 2, 3 and 4 hold. Then, for any fixed $k$ with $k^* \le k \le d$, the I-OBS algorithm has the following local quadratic convergence rate:*

$$\|\theta_{t+1} - \theta^*\|_2 \le \left(1 + \sqrt{\frac{L}{\mu} \frac{d-k^*}{d-k}}\right) \frac{M}{2\mu} \|\theta_t - \theta^*\|_2^2. \tag{9}$$

The quadratic convergence rate in the form $\|\theta_{t+1} - \theta^*\|_2 \le c\|\theta_t - \theta^*\|_2^2$ implies a $\mathcal{O}(\log\log\frac{1}{\epsilon})$ iteration complexity for achieving an $\epsilon$-error with initialization $\|\theta_0 - \theta^*\|_2 \le \frac{1}{2c}$. This complexity matches the classical Newton's method up to constants. Due to space constraints, we defer the complete proof to Appendix B.4. Here, we provide a brief overview of the proof idea.

*Proof sketch.* We split the error $\theta_{t+1} - \theta^*$ into two parts: the standard error $\theta_t - \mathbf{H}_t^{-1}\nabla f(\theta_t) - \theta^*$ corresponding to the Newton's method and an additional error $\mathbf{H}_t^{-1}\mathbf{H}_t^{S^{t+1}}\left(\theta_t - \mathbf{H}_t^{-1}\nabla f(\theta_t)\right)$ stemming from the sparsity. The key technical challenge of the proof is to bound the second error by the first one, which then can be bounded by the squared error $\|\theta_t - \theta^*\|_2^2$ using standard tools.

Notice that the matrix $\mathbf{H}_t^{-1}\mathbf{H}_t^{S^{t+1}}$ of the second error term is a projection matrix (see Lemma 7). This observation allows bounding the error by $\|\theta_t - \mathbf{H}_t^{-1}\nabla f(\theta_t)\|_2$, which however, may not converge to zero as $\theta^*$ is missing. To get a tighter upper bound, we utilize the structure of $\mathbf{H}_t^S$ and the definition of $S^{t+1}$ to show

$$\|\mathbf{H}_t^{-1}\mathbf{H}_t^{S^{t+1}}\theta_t^+\|_2 \leq \sqrt{\tfrac{L}{\mu}}\|\mathbf{I}_{\tilde{S}}\theta_t^+\|_2 \tag{10}$$

for any set $\tilde{S} \subseteq [d]$ with $|\tilde{S}| = |S^{t+1}| = d - k$, where $\theta_t^+ = \theta_t - \mathbf{H}_t^{-1}\nabla f(\theta_t)$. Then, we choose the set $\tilde{S}$ to minimize the norm in the right, namely $\tilde{S} = [d] \setminus \operatorname{supp} \theta_t^+$. With this choice of the set $\tilde{S}$ we invoke Lemma 8 and get

$$\|\mathbf{I}_{\tilde{S}}\theta_t^+\|_2 = \|T_k(\theta_t^+) - \theta_t^+\|_2 \leq \sqrt{\tfrac{d-k^*}{d-k}}\|\theta_t^+ - \theta^*\|_2. \tag{11}$$

Combining (10) and (11) together with the bound $\|\theta_t^+ - \theta^*\|_2 \leq \tfrac{M}{2\mu}\|\theta_t - \theta^*\|_2^2$ of the standard Newton's method concludes the proof of our main claim in Eqn. (9). $\qquad\square$

**Discussion and Implications.** Although it offers strong convergence guarantees, the vanilla version of I-OBS is hard to implement in practice. The difficulties are two-fold. First, in computing $\mathbf{H}_t^S$, we need to compute $(\mathbf{I}_S\mathbf{H}_t^{-1}\mathbf{I}_S^\top)^{-1}$ which is the inverse of a $|S| \times |S|$ matrix, requiring $\mathcal{O}(|S|^3)$ complexity in general. Also, if $|S| = cd$ for some $c \in (0,1)$, i.e. we prune a $c$-fraction of weight, then the complexity of this step will be $\mathcal{O}(d^3)$. Second, to obtain the optimal set $S^{t+1}$, we need to solve a combinatorial problem $\arg\min_{S:\,|S|=d-k}(\theta_t^+)^\top\mathbf{H}_t^S\theta_t^+$. This problem is NP-hard [Chen et al., 2014, Natarajan, 1995], and brute-forcing over all possible sets has the complexity of $\mathcal{O}(d^k)$. If $k = cd$, then the complexity will be $\Omega(e^d)$ which is too expensive in practical applications. Thus, we propose a practical variants of I-OBS which corresponds to Algorithm 1 (Option 1), namely Top$k$-I-OBS. We also provide local convergence guarantee for Top$k$-I-OBS. Besides, we also proposed another version called stochastic I-OBS in Appendix 3.

**3.6. A Practical Variant: Top$k$-OBS.** As mentioned above, one natural implementation of the I-OBS method in practice is to replace the multiplication by the matrix $\mathbf{I} - \mathbf{H}_t^{-1}\mathbf{H}_t^{S^{t+1}}$ with directly applying the Top-$k$ operator to the preconditioned gradient step. This leads to the following Top$k$-I-OBS update corresponding to Algorithm 1 with Option 1:

$$\theta_{t+1} = T_k(\theta_t - \mathbf{H}_t^{-1}\nabla f(\theta_t)) \tag{12}$$

For the Top$k$-I-OBS method, we can still show local convergence similar to Theorem 1.

**Lemma 3.** *Let Assumptions 1, 2, 3 and 4 hold. Then I-OBS method defined in (12) has the following convergence rate:*

$$\|\theta_{t+1} - \theta_*\|_2 \leq \left(1 + \sqrt{\tfrac{k^*}{k}}\right)\tfrac{M}{2\mu}\|\theta_t - \theta^*\|_2^2 \tag{13}$$

The proof is deferred to Appendix B.5. We remark that similar algorithms are studied for sparse linear regression problems with a linear global rate, while our Top$k$-I-OBS method applies to a wider class of problems, and obtains a quadratic local rate.

# 4 Experiments

## 4.1 Synthetic Experiments for Sparse Linear Regression

To validate our theoretical analysis, we first consider the sparse linear regression problem with two different priors. First, we consider a sparse linear regression problem with Gaussian prior. In particular, we first sample a signal $\theta_* \in \mathbb{R}^d$ of dimension $d = 128$ from a standard Gaussian prior, and then we set it to be sparse by uniformly random keeping $k_* = 16$ entries, and set the rest to zero. Then, we generate $n = 256$ sensing vectors $\boldsymbol{X} = [\boldsymbol{x}_1, \boldsymbol{x}_2, \dots, \boldsymbol{x}_n]^\top \in \mathbb{R}^{n \times d}$ i.i.d. from a standard Gaussian distribution with variance $\tfrac{1}{n}$, and the label is generated via a linear measurement $y_i = \langle \theta_*, \boldsymbol{x}_i \rangle$. We aim to recover the signal by minimizing the empirical mean square loss $\mathcal{L}(\theta) = \|\boldsymbol{y} - \boldsymbol{X}\theta\|_2^2$, and set the sparsity for each step to $k = 64$. It is easy to see that in this case, the Hessian of the loss function is $\boldsymbol{H} := \nabla^2\mathcal{L}(\theta) = \boldsymbol{X}^\top\boldsymbol{X}$, and this matrix is with high probability positive definite.

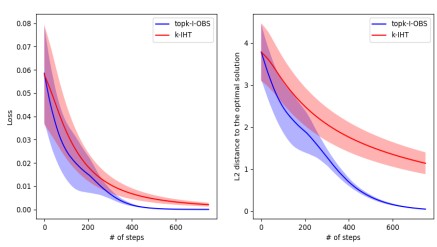


(a) Learning curve with standard Gaussian prior.

(b) Recovery of the MNIST prior.

Figure 1: Comparison of $k$-IHT and Top$k$-WoodTaylor for sparse linear regression with standard Gaussian and MNIST priors.

We compare $k$-IHT [Peste et al., 2021] and Top$k$-I-OBS (12) numerically. For $k$-IHT, we apply the learning rate $\eta = \frac{1}{\lambda_{\max}(H)}$. We run each algorithm for $750$ steps, and run $20$ independent experiments then take the average over all the experiments. The synthetic experiments can be done on a personal laptop. The results are shown in Figure 1a. We see that the top$k$-WoodTaylor method indeed has a better convergence rate compared to $k$-IHT due to the use of second-order information in Newton's step. In particular, the left plot of Figure 1a characterizes the training loss, and the right plot characterizes the distance to the optimal solution.

In the second setting, we let the signal be uniformly sampled from the MNIST dataset. In particular, we randomly sample an image of MNIST training dataset and use this image as the signal $\theta_*$. In this case, the dimension $d$ of the signal is $784$, and we denote the sparsity of the signal as $k_*$. Next we generate $n = 2d = 1568$ sensing vectors $X$ i.i.d from a standard Gaussian distribution with variance $\frac{1}{n}$, and the label is generated via a linear measurement $y_i = \langle \theta_*, x_i \rangle$. Similarly, we minimize the MSE loss, and the sparsity we set for each step is $k = 2k_*$. We also compare $k$-IHT and Top$k$-I-OBS numerically. We run each algorithm for $4000$ steps and run $20$ independent experiments. For the $k$-IHT, similarly, we apply the maximum possible learning rate.

We show the recovered signals in Figure 1b. (The learning curve is almost identical to the Gaussian case and therefore omitted.) When plotting the recovered image, we only plot the value between $[0, 255]$. We can see from the learning curve that the Top$k$-I-OBS has a better convergence rate, which results in a better recovery quality compared to $k$-IHT for a fixed number of steps.

## 4.2 Applying I-OBS to Model Pruning

In this section we describe our implementations for selecting the set of weights to prune using second order information applied to practical models.

---

**Algorithm 2** Iterative Optimal Brain Surgeon (I-OBS) for practical model pruning

---

1: **Input:** Sparsity threshold $k_\ell \in [d]$ for each layer $\ell \in \{1, 2, \dots, L\}$
2: **for** each round $t \in \{1, 2, \dots, T\}$ **do**
3:     Sample a data batch $X_t$
4:     ▷ *Pruning and optimization step:*
5:     $H^0 \leftarrow X_t$
6:     **for** each layer $\ell \in \{1, 2, \dots, L\}$ **do**
7:         Solve the constrained optimization problem

$$\min_{\widehat{W}^\ell} \|W_{t-1}^\ell H^{\ell-1} - \widehat{W}^\ell H^{\ell-1}\|_2^2 \quad s.t. \ \|\widehat{W}^\ell\|_0 = k_\ell$$

        using a quadratic sparsity solver such as OBC or SparseGPT.
8:         $W_{t-1}^\ell \leftarrow \widehat{W}^\ell$
9:         $W_t^\ell \leftarrow W_{t-1}^\ell - \eta g_t^\ell(X_t, W_{t-1}^\ell)$ for each $\ell \in \{1, 2, \dots, L\}$
10:     **end for**
11: **end for**

---

**Detailed algorithm.** The algorithms we implement for pruning are described in Algorithm 2. We consider pruning an $L$-layer feedforward neural network $f(X; W_1, \ldots W_L) = \sigma_L \circ W^L \circ \cdots \circ \sigma_\ell \circ W^\ell \circ \ldots \sigma_1 \circ W^1 X$, but a similar pruning strategy would apply to neural networks with different structure. More specifically, we consider a pretrained model, and we aim to find a sparse model using the I-OBS algorithm. We apply a layer-wise pruning strategy as in OBC [Frantar and Alistarh, 2022]. In particular, for each round $t$, we sample a new data batch $X_t$, and solve the quadratic sparse optimization problem

$$\min_{\widehat{W}} \|WX - \widehat{W}X\|_2^2 \quad s.t. \ \|\widehat{W}\|_0 = k$$

where $H^\ell = \sigma_{\ell-1} \circ W^{\ell-1} \circ \cdots \circ \sigma_1 \circ W^1 X$ is the input of the $\ell$-th layer. We use the SparseGPT method [Frantar and Alistarh, 2022] as an approximate solver for this problem. The connection between one-shot pruners such as OBC or SparseGPT and I-OBS was discussed in Section 3.

**Example 1: Pruning Vision Transformers (ViTs).** We now apply this approach to pruning DNNs from the ViT family [Dosovitskiy et al., 2021]. For this, we instantiate the above pruning framework for the case of the SparseGPT solver in the context of I-OBS, to prune the DeiT-Tiny (5M parameters), DeiT-Small (22M parameters) and DeiT-Base (86M parameters) models in one shot to 50% uniform sparsity, starting from an accurate pretrained model, using 1000 ImageNet calibration samples. We present our results in Table 1, where the single-iteration result is given by the one-shot algorithm, which in this instance is SparseGPT.

Table 1: Pruning results for ViTs (DeiT) using SparseGPT. We report Top-1 accuracy results on the ImageNet validation set.

| Model | # iterations | | | | | | |
|---|---|---|---|---|---|---|---|
| | D | 1 | 10 | 25 | 50 | 75 | 100 |
| **DeiT-Tiny** | 72.08 | 62.98 | 63.61 | 64.01 | 64.11 | 64.13 | 64.05 |
| **DeiT-Small** | 79.81 | 76.07 | 76.28 | 76.36 | 76.50 | 76.57 | 76.57 |
| **DeiT-Base** | 81.91 | 80.29 | 80.30 | 80.37 | 80.42 | 80.46 | 80.43 |

The results show that I-OBS can bring consistent improvements across all three model sizes, with the largest improvements coming at the smallest model size (5M), where the performance drop relative to the dense model is most pronounced. We also observe that improvements quickly saturate at the larger model sizes and with respect to increasing the number of I-OBS iterations, suggesting that the sparse solution is already close to the local optimum in the case of highly-overparametrized models.

**Example 2: Pruning Large Language Models (LLMs).** Finally, we extend the previous approach to LLMs, again comparing against SparseGPT as a baseline. Specifically, we estimate the Hessian matrix at each layer by using 128 calibration samples from the WikiText2 dataset for OPT-125M and from Red Pajama dataset for Phi-1.5B, respectively. We report the accuracy in terms of perplexity (lower is better) on the WikiText2 and C4 datasets. In this scenario, we start from a dense model and prune it to 50% uniform sparsity, using I-OBS for 3 iterations. (We observed that using a larger number of iterations leads to "overfitting," by which we mean good PPL on WikiText2 but worsening on C4.) The results are given in Table 2.

Table 2: Pruning results for Phi-1.5M using SparseGPT. We report perplexity (the lower, the better).

| Model | # samples | WikiText2 | | | C4 | | |
|---|---|---|---|---|---|---|---|
| | | Dense | SparseGPT | I-OBS(3) | Dense | SparseGPT | I-OBS(3) |
| **OPT-125M** | 128 | 27.65 | 33.85 | 25.20 | 24.61 | 32.27 | 31.41 |
| **Phi-1.5** | 128 | 21.82 | 25.28 | 23.94 | 20.90 | 21.13 | 20.26 |

In both cases, we observe that I-OBS leads to significant improvements in terms of perplexity, specifically of approximately one PPL point on C4, relative to SparseGPT. This effect confirms the fact that I-OBS is significantly better at minimizing the per-layer compression loss relative to a single iteration of SparseGPT.

We also applied our methods on Llama-2(7B) and Llama-3(8B) models, and the results are discussed in Appendix A.2

## 5   Limitations and Future works

In this section, we discuss the limitation of this work, and address a few directions for future work.

From the theoretical perspective, our main limitation is the mild gap between the analytical assumptions and the practical setting we considered in the pruning experiments. We mention that our assumptions are standard in the literature (see e.g. [Peste et al., 2021, Analytical Assumptions]) to guarantee the convergence of gradient-based methods, it does not hold in general for practical problem such as model pruning. Thus, one interesting direction for future work is to relax those assumptions. In addition, in Theorem 1 and Lemma 3 we only provide local convergence rate, which mean that we require the initial distance from the global optimum to be small enough. While such local convergence results are typical for Newton type algorithms (see e.g. [Nesterov, 2018, Theorem 1.2.5]), there are a few modifications of the vanilla Newton's methods which gives a global convergence rate such as adding a cubic regularization term [Nesterov, 2018, Section 4.1] or adding a proper adaptive quadratic regularization term [Mishchenko, 2023]. The challenge is how to incorporate the sparsity constrain to make each step sparse, and we leave this direction for future works. For the experiments, it would be interesting to apply the pruning method to larger models, which we plan to do in future work.

## Acknowledgements

The authors thank the anonymous NeurIPS reviewers for their useful comments and feedback, the IT department from the Institute of Science and Technology Austria for the hardware support, and Weights and Biases for the infrastructure to track all our experiments. Mher Safaryan has received funding from the European Union's Horizon 2020 research and innovation program under the Maria Skłodowska-Curie grant agreement No 101034413.

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

# Contents

# A  Experimental Details

In this section we provide details about the hyper-parameters that we used for both experiments described in Section 4.

## A.1  I-OBS for Model Pruning

**Pruning ViTs.** We use ViTs from the `timm` package that are pretrained on ImageNet using RGB images of size $224 \times 224$. We set both learning rate and hessian dampening to $0.01$, sparsity to $50\%$ and prune the query, key, values, projection and fully connected layers for $100$ iterations using batch size $128$.

**Pruning LLMs.** We use the pretrained OPT-125m model from Meta in *bfloat16* format with sequence length $2048$ and prune it to $50\%$ uniform sparsity using $3$ iterations of I-OBS on batches of size $128$ (calibration samples) from the WikiText2 using both learning rate and hessian dampening set to $0.01$.

We use the pretrained Phi-1.5 model from Microsoft also in *bfloat16* format with sequence length $2048$ and prune it to $50\%$ uniform sparsity using $3$ iterations of I-OBS on batches of size $128$ (calibration samples) from the Red Pajama dataset using learning rate $1e - 5$ and hessian dampening set to $0.01$.

## A.2 Extra experiments

**I-OBS for LLMs** To study the scalability of I-OBS pruning to LLM (Large Language Models) we consider iterative pruning followed by finetuning for Llama-2 (7B) and Llama-3 (8B) models. Specifically, we apply SparseGPT [Frantar and Alistarh, 2023] pruner with Hessians estimated using 8M tokens (2k sequences of length 4096 for Llama-2 and 1k sequences of length 8192 for Llama-3) and use the same calibration set for finetuning [2]. After each pruning step we tune the model for 1 epoch *without* applying any sparsity mask, i.e. all parameters may become non-zero before some of them are projected back on next pruning iteration. We adopt Adam optimizer with learning rate of $10^{-5}$, batch size of 16 for Llama-2 and 8 for Llama-3. The dynamics of evaluation metrics for both model is shown on Figures 2 and 3. We also evaluate the pruned model on MMLU tasks, and the results are shown in the following Table 3 and 4

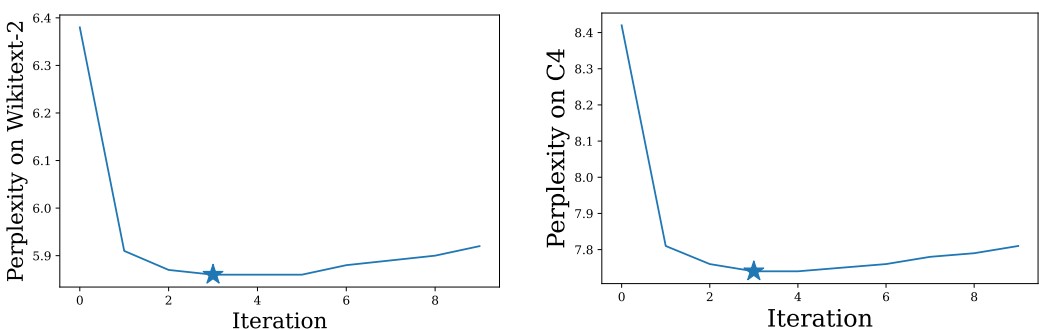

Figure 2: I-OBS dynamics for Llama-2 7B (star corresponds to best validation score). (**Left**) Wikitext-2 Perplexity vs iteration. (**Right**) C4 Perplexity vs iteration.

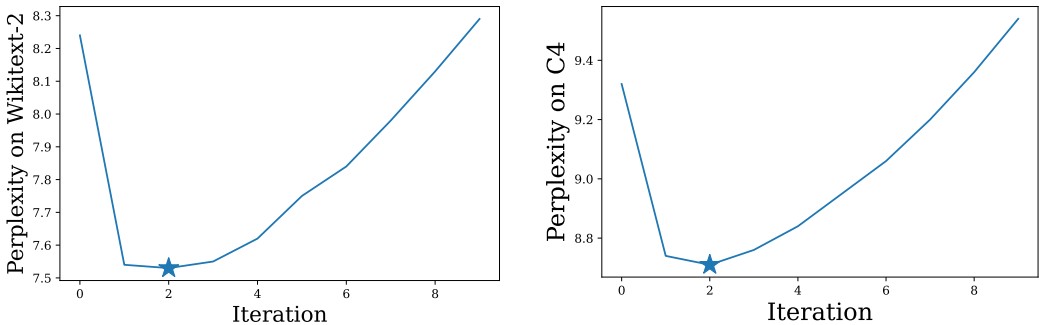

Figure 3: I-OBS dynamics for Llama-3 8B (star corresponds to best validation score). (**Left**) Wikitext-2 Perplexity vs iteration. (**Right**) C4 Perplexity vs iteration.

One can observe that the quality of sparse solution significantly improves after first finetuning iteration. Then there is small improvement during the next few iterations, followed by gradual performance deterioration afterward. We speculate that this behavior is due to the overfitting on the calibration set.

**Comparing with CBS** In Table 5 of the attached file in global rebuttal, we provide results for I-OBS applied to MobileNetV1 model used in STR, which is the same model studied in Table 4 of [Yu et al., 2022]. We skip the depth-wise convolutions having shape $(C, 1, K, K)$ when we apply our pruning algorithm, as is standard. Starting with 60% sparsity, our I-OBS pruner outperforms CBS method on MobileNetV1 by a large margin: 2% for 60% sparsity, 5% for 70% sparsity and 12% for 80% sparsity. For low sparsities (30% to 50%), the two methods are comparable, since the accuracy difference is less than 0.5%.

---

[2]Calibration data is sampled from Red Pajama dataset.

Table 3: Performance of I-OBS on Llama-2-7b

| Iterations | MMLU(5-shot) |
|------------|--------------|
| 0 (dense)  | 0.4584 |
| 1          | 0.3878 |
| 2          | **0.3950** |
| 3          | 0.3932 |
| 4          | 0.3943 |
| 5          | 0.3946 |
| 6          | 0.3929 |
| 7          | 0.3919 |
| 8          | 0.3893 |
| 9          | 0.3903 |
| 10         | 0.3863 |

Table 4: Performance of I-OBS on Llama-3-8B

| Iterations | MMLU(5-shot) |
|------------|--------------|
| 0 (dense)  | 0.6525 |
| 1          | 0.5476 |
| 2          | 0.5577 |
| 3          | 0.5582 |
| 4          | **0.5605** |
| 5          | 0.5553 |
| 6          | 0.5521 |
| 7          | 0.5444 |
| 8          | 0.5449 |
| 9          | 0.5403 |
| 10         | 0.5375 |

Table 5: Comparison of I-OBS and CBS at different sparsity levels

| sparsity | I-OBS | CBS |
|----------|-------|-----|
| 30%      | 71.6  | **71.88** |
| 40%      | 71.3  | **71.45** |
| 50%      | **70.6** | 70.21 |
| 60%      | **68.4** | 66.37 |
| 70%      | **60.7** | 55.11 |
| 80%      | **27.9** | 16.38 |
| 0% (dense) | 71.95% | |

# B Deferred Proofs

## B.1 Stochastic I-OBS

To enhance the computational efficiency of the I-OBS approach (8) to large scale problems, we introduce certain relaxations to approximate the full-batch gradient $\nabla f(\theta_t)$ and the Hessian $\mathbf{H}_t$.

Suppose we have access to stochastic gradients $g(\theta)$ that is an unbiased estimator of the full-batch gradient, namely $\mathbb{E}[g(\theta)] = \nabla f(\theta)$. Inspired by the Fisher approximation [Amari, 1998], we over-approximate the Hessian using the outer products of stochastic gradients as follows:

$$\nabla^2 f(\theta) \preceq \lambda \mathbf{I} + \mathbb{E}[\mathbf{G}(\theta)], \text{ with } \mathbf{G}(\theta) = \frac{g(\theta)g(\theta)^\top}{\|g(\theta)\|_2}, \tag{14}$$

where $\lambda \geq 0$ and for two matrices $\mathbf{A}, \mathbf{B}$ the inequality $\mathbf{A} \preceq \mathbf{B}$ means that $\mathbf{B} - \mathbf{A}$ is positive semi-definite. Compared to the Fisher approximation we scaled the second term inside the expectation by divided with the norm $\|g(\theta)\|_2$ to avoid scaling issues. Since the stochastic matrix composed of outer

product inside the expectation is positive semi-definite, the condition (14) holds automatically under $L$-smoothness assumption with $\lambda = L$.

For example, if we use full-batch gradient $g(\theta) = \nabla f(\theta)$, then the matrix $\mathbb{E}[\mathbf{G}(\theta)]$ is of rank 1. In this case $\lambda$ should be set to the largest eigenvalue $\lambda_{\max}(\mathbf{H}_t)$ of the Hessian. However, when batch size of $g(\theta)$ is smaller, then the expectation of stochastic rank-1 matrices $\mathbf{G}(\theta)$ would potentially be of full-rank, allowing us to choose a smaller value for $\lambda$.

An alternative motivation behind condition (14) is given by empirical observations on the correlation between loss smoothness and gradient norm [Zhang et al., 2020, Wang et al., 2022]. Since $\lambda_{\max}(\mathbf{G}(\theta)) = \|g(\theta)\|_2$ and $\|\nabla f(\theta)\| \leq \mathbb{E}[\|g(\theta)\|]$, condition (14) can be viewed as matrix version of $(L_0, L_1)$-smoothness condition by Zhang et al. [2020]:

$$\|\nabla^2 f(\theta)\|_2 \leq L_0 + L_1 \|\nabla f(\theta)\|_2 \tag{15}$$

allowing the smoothness to grow with the gradient norm.

Applying the condition (14), we can over-approximate the quadratic objective of (7) by $\mathbb{E}[\langle g(\theta_t), \theta - \theta_t \rangle + \frac{|\langle g(\theta_t), \theta - \theta_t \rangle|^2}{2\|g(\theta_t)\|_2} + \frac{\lambda}{2}\|\theta - \theta_t\|^2]$. The stochastic quadratic problem of the latter objective can be solved analytically if we specify the sparsity constraint.

**Lemma 4.** *With fixed sparsity constraint $\mathbf{I}_S \theta = 0$, the stochastic quadratic problem*

$$\arg\min_{\theta : \mathbf{I}_S \theta = 0} \langle g(\theta_t), \theta - \theta_t \rangle + \frac{|\langle g(\theta_t), \theta - \theta_t \rangle|^2}{2\|g(\theta_t)\|_2} + \frac{\lambda}{2}\|\theta - \theta_t\|^2$$

*has the following solution for the update*

$$\theta_{t+1} = \mathbf{E}_Q \left( \theta_t - \frac{1}{\lambda + \|\mathbf{E}_Q g(\theta_t)\|_2^2 / \|g(\theta_t)\|_2} g(\theta_t) \right),$$

*where the mask $Q$ is the complement of $S$.*

In general, the closed form solution for the mask $Q$ is not tractable. For this reason, we propose the following stochastic version of I-OBS:

$$\theta_{t+1} = T_k \left( \theta_t - \frac{1}{\lambda + \|\mathbf{E}_{Q^t} g(\theta_t)\|_2^2 / \|g(\theta_t)\|_2} g(\theta_t) \right),$$

with $Q^t = \operatorname{supp} \theta_t$.

## B.2 Proof of Lemma 1

**Lemma 1.** *The closed-form solution to (6) with linear model $\phi_t(\theta) = f(\theta_t) + \langle \nabla f(\theta_t), \theta - \theta_t \rangle$ is*

$$\theta_{t+1} = T_k(\theta_t - \eta \nabla f(\theta_t)).$$

*Proof of Lemma 1.* Plugging the linear model $\phi_t(\theta) = f(\theta_t) + \langle \nabla f(\theta_t), \theta - \theta_t \rangle$ into the problem (6) and dropping the constant term $f(\theta_t)$ in the $\arg\min$, we get the following problem:

$$\arg\min_{\theta : \|\theta\|_0 \leq k} \left\{ \langle \nabla f(\theta_t), \theta - \theta_t \rangle + \frac{1}{2\eta} \|\theta - \theta_t\|_2^2 \right\}. \tag{16}$$

We break this problem into two sub-problems: first we solve the problem with fixed sparsity mask, then optimize over the mask. Formally, for a given mask $Q \subseteq [d]$ with cardinality $|Q| = k$, we solve

$$\theta_{t+1}^Q = \arg\min_{\theta : \mathbf{I}_S \theta = 0} \left\{ \langle \nabla f(\theta_t), \theta - \theta_t \rangle + \frac{1}{2\eta} \|\theta - \theta_t\|_2^2 \right\}, \tag{17}$$

where $S = [d] \setminus Q$ is the complement of the mask, i.e., the coordinates that we want to drop. Afterwards, we find the optimal mast $Q^{t+1}$ by solving

$$Q^{t+1} = \arg\min_{Q : |Q| = k} \left\{ \langle \nabla f(\theta_t), \theta_{t+1}^Q - \theta_t \rangle + \frac{1}{2\eta} \|\theta_{t+1}^Q - \theta_t\|_2^2 \right\}. \tag{18}$$

Then, the solution to (16) would be $\theta_{t+1} = \theta_{t+1}^{Q^{t+1}}$.

Let us start solving the first sub-problem in (17). As the problem is convex (convex quadratic objective with linear constraints), we can solve it with the method of Lagrange multipliers. With a modified primal variable $\delta = \theta - \theta_t \in \mathbb{R}^d$ and dual variable $\lambda \in \mathbb{R}^{|S|}$ the Lagrangian takes the form

$$L(\delta, \lambda) = \langle \nabla f(\theta_t), \delta \rangle + \frac{1}{2\eta} \|\delta\|_2^2 + \lambda^\top (\mathbf{I}_S(\delta + \theta_t)).$$

Applying Karush–Kuhn–Tucker (KKT) conditions to our problem yields

$$0 = \nabla_\delta L(\delta^*, \lambda^*) = \nabla f(\theta_t) + \frac{1}{\eta}\delta^* + \mathbf{I}_S^\top \lambda$$

$$0 = \nabla_\lambda L(\delta^*, \lambda^*) = \mathbf{I}_S(\delta^* + \theta_t).$$

Solving the first equation for $\delta$ variable, we get $\delta^* = -\eta(\nabla f(\theta_t) + \mathbf{I}_S^\top \lambda^*)$. Plugging this into the second equation gives

$$0 = \mathbf{I}_S(-\eta(\nabla f(\theta_t) + \mathbf{I}_S^\top \lambda^*) + \theta_t) = \mathbf{I}_S(\theta_t - \eta\nabla f(\theta_t)) - \eta \mathbf{I}_S \mathbf{I}_S^\top \lambda^*.$$

Noticing that $\mathbf{I}_S \mathbf{I}_S^\top$ is the identity matrix in $\mathbb{R}^{|S| \times |S|}$ and $\mathbf{I}_S^\top \mathbf{I}_S = \mathbf{E}_S$, we conclude

$$\lambda^* = \frac{1}{\eta}\mathbf{I}_S(\theta_t - \eta\nabla f(\theta_t)),$$

$$\delta^* = -\eta\nabla f(\theta_t) - \mathbf{E}_S(\theta_t - \eta\nabla f(\theta_t)),$$

which implies that the solution to our first sup-problem (17) can be given as

$$\theta_{t+1}^Q = \delta^* + \theta_t$$
$$= \theta_t - \eta\nabla f(\theta_t) - \mathbf{E}_S(\theta_t - \eta\nabla f(\theta_t))$$
$$= (\mathbf{I} - \mathbf{E}_S)(\theta_t - \eta\nabla f(\theta_t)) = \mathbf{E}_Q(\theta_t - \eta\nabla f(\theta_t)).$$

To solve the second sub-problem (18), we plug in the obtained solution $\theta_{t+1}^Q$ and simplify the objective function as follows:

$$\langle \nabla f(\theta_t), \theta_{t+1}^Q - \theta_t \rangle + \frac{1}{2\eta} \|\theta_{t+1}^Q - \theta_t\|_2^2 = \frac{1}{2\eta}\left( \|\theta_{t+1}^Q - \theta_t + \eta\nabla f(\theta_t)\|_2^2 - \|\eta\nabla f(\theta_t)\|_2^2 \right)$$

$$= \frac{1}{2\eta}\|(\mathbf{I} - \mathbf{E}_Q)(\theta_t - \eta\nabla f(\theta_t))\|_2^2 - \frac{\eta}{2}\|\nabla f(\theta_t)\|_2^2$$

$$= \frac{1}{2\eta}\|\mathbf{E}_S(\theta_t - \eta\nabla f(\theta_t))\|_2^2 - \frac{\eta}{2}\|\nabla f(\theta_t)\|_2^2.$$

Recall that we need to minimize the obtained expression with respect to the mask $Q$, or equivalently its complement $S$. As the second term, $-\frac{\eta}{2}\|\nabla f(\theta_t)\|_2^2$, does not depend on the mask, we can ignore it. To minimize the first term, $\|\mathbf{E}_S(\theta_t - \eta\nabla f(\theta_t))\|_2^2$, we need to choose the projection set $S$ corresponding to the coordinates of the vector $\theta_t - \eta\nabla f(\theta_t)$ with minimal absolute values. In other words, the mask $Q$ should be chosen to be the coordinates corresponding to the largest magnitudes of the vector, namely $Q^{t+1} = \mathrm{supp}\left(T_k(\theta_t - \eta\nabla f(\theta_t))\right)$.

Thus, combining the solutions of these two problems we conclude that $\theta_{t+1} = T_k(\theta_t - \eta\nabla f(\theta_t))$. $\quad\square$

### B.3 Proof of Lemma 2

**Lemma 2.** *If $\mathbf{H}_t$ is positive definite, then each step of Algorithm 1 (Option 2) solves Eqn.(8).*

*Proof of Lemma 2.* To solve the optimization problem in (8), we break this problem into two parts. First, we fix subset $Q \subseteq [d]$ with size $k$ and complement $S = [d] \setminus Q$, then we find the optimal set $Q$ (or equivalently $S$). The first step is to solve the following quadratic optimization problem:

$$\theta_{t+1}^S = \arg\min_{\theta : \mathbf{I}_S\theta = 0} \langle \nabla f(\theta_t), \theta - \theta_t \rangle + \frac{1}{2}\langle \theta - \theta_t, \mathbf{H}_t(\theta - \theta_t) \rangle, \tag{19}$$

Next, we aim to find the optimal set to prune, which is given by

$$S^{t+1} = \underset{S\,:\,|S|=d-k}{\arg\min}\ \langle \nabla f(\theta_t), \theta_{t+1}^S - \theta_t \rangle + \frac{1}{2}\langle \theta_{t+1}^S - \theta_t, \mathbf{H}_t(\theta_{t+1}^S - \theta_t)\rangle. \tag{20}$$

Once we solve both the problems (19) and (20), we set $\theta_{t+1} = \theta_{t+1}^{S^{t+1}}$.

To solve (19), notice that the optimization problem is convex with $S$ linear equality constraints. Therefore we can solve it via KKT conditions. With a change of variable $\delta = \theta - \theta_t \in \mathbb{R}^d$, we define the Lagrangian with multipliers $\lambda \in \mathbb{R}^{|S|}$ as follows:

$$L(\delta, \lambda) = \langle \nabla f(\theta_t), \delta \rangle + \frac{1}{2}\langle \delta, \mathbf{H}_t\delta \rangle + \langle \theta_t + \delta, \mathbf{I}_S^\top \lambda\rangle.$$

Then the KKT conditions become

$$0 = \nabla_\delta L(\delta^*, \lambda^*) = \nabla f(\theta_t) + \mathbf{H}_t\delta^* + \mathbf{I}_S^\top\lambda^*.$$
$$0 = \nabla_\lambda L(\delta^*, \lambda^*) = \mathbf{I}_S(\delta^* + \theta_t).$$

Note that the matrix $\mathbf{H}_t$ invertible since it is positive definite. From the first equation we obtain that $\delta^* = -\mathbf{H}_t^{-1}(\nabla f(\theta_t) + \mathbf{I}_S^\top\lambda^*)$. Then, plugging this into the second equation, we get $(\mathbf{I}_S\mathbf{H}_t^{-1}\mathbf{I}_S^\top)\lambda^* = \mathbf{I}_S(\theta_t - \mathbf{H}_t^{-1}\nabla f(\theta_t))$. Note that $\mathbf{I}_S\mathbf{H}_t^{-1}\mathbf{I}_S^\top$ is basically the matrix $\mathbf{H}_t^{-1}$ removing the rows and columns outside $S$.

Next, we claim that the matrix $\mathbf{I}_S\mathbf{H}_t^{-1}\mathbf{I}_S^\top$ is also positive definite. Indeed, by the interlacing property in Lemma 9, we have:

$$0 < \frac{1}{\lambda_d(\mathbf{H}_t)} \le \lambda_1(\mathbf{H}_t^{-1}) \le \lambda_1(\mathbf{I}_S\mathbf{H}_t^{-1}\mathbf{I}_S^\top),$$

which implies that all eigenvalues of $\mathbf{I}_S\mathbf{H}_t^{-1}\mathbf{I}_S^\top$ are positive, hence it is also invertibility. Therefore, we get

$$\lambda^* = \left(\mathbf{I}_S\mathbf{H}_t^{-1}\mathbf{I}_S^\top\right)^{-1}\mathbf{I}_S(\theta_t - \mathbf{H}_t^{-1}\nabla f(\theta_t)) =: \lambda_{t,S} \tag{21}$$
$$\delta^* = -\mathbf{H}_t^{-1}\nabla f(\theta_t) - \mathbf{H}_t^{-1}\mathbf{I}_S^\top\lambda^* =: \delta_{t,S}. \tag{22}$$

Finally, we have the following update rule for I-OBS:

$$\theta_{t+1}^S = \theta_t + \delta_{t,S} = \left(\mathbf{I} - \mathbf{H}_t^{-1}\mathbf{H}_t^S\right)\left(\theta_t - \mathbf{H}_t^{-1}\nabla f(\theta_t)\right), \tag{23}$$

where $\mathbf{H}_t^S = \mathbf{I}_S^\top\left(\mathbf{I}_S\mathbf{H}_t^{-1}\mathbf{I}_S^\top\right)^{-1}\mathbf{I}_S \in \mathbb{R}^{d\times d}$. Next, we should pick the optimal $S^* = S^{t+1}$ via (20), that is, we let:

$$S^{t+1} = \underset{S\subseteq[d],|S|=d-k}{\arg\min}\ \langle \nabla f(\theta_t), \theta_{t+1}^S - \theta_t \rangle + \frac{1}{2}\langle \theta_{t+1}^S - \theta_t, \mathbf{H}_t(\theta_{t+1}^S - \theta_t)\rangle$$

Plugging the expression (23) of $\theta_{t+1}^S$ into the above definition of $S^{t+1}$, and ignoring terms that do not depend on $S$, we have

$$
\begin{aligned}
S^{t+1} &= \underset{S\,:\,|S|=d-k}{\arg\min}\ \langle \nabla f(\theta_t), \theta_{t+1}^S - \theta_t \rangle + \frac{1}{2}\langle \theta_{t+1}^S - \theta_t, \mathbf{H}_t(\theta_{t+1}^S - \theta_t)\rangle \\
&= \underset{S\,:\,|S|=d-k}{\arg\min}\ \langle \nabla f(\theta_t), \theta_{t+1}^S \rangle + \frac{1}{2}\langle \theta_{t+1}^S, \mathbf{H}_t\theta_{t+1}^S\rangle - \langle \theta_t, \mathbf{H}_t\theta_{t+1}^S\rangle \\
&= \underset{S\,:\,|S|=d-k}{\arg\min}\ -\langle \mathbf{H}_t\theta_{t+1}^S, \theta_t - \mathbf{H}_t^{-1}\nabla f(\theta_t)\rangle + \frac{1}{2}\langle \theta_{t+1}^S, \mathbf{H}_t\theta_{t+1}^S\rangle \\
&= \underset{S\,:\,|S|=d-k}{\arg\min}\ -\langle (\mathbf{H}_t - \mathbf{H}_t^S)(\theta_t - \mathbf{H}_t^{-1}\nabla f(\theta_t)), \theta_t - \mathbf{H}_t^{-1}\nabla f(\theta_t)\rangle \\
&= \underset{S\,:\,|S|=d-k}{\arg\min}\ (\theta_t - \mathbf{H}_t^{-1}\nabla f(\theta_t))^\top\mathbf{H}_t^S(\theta_t - \mathbf{H}_t^{-1}\nabla f(\theta_t)). \tag{24}
\end{aligned}
$$

Finally, setting $\theta_{t+1} = \theta_{t+1}^{S^{t+1}}$ finishes the proof. $\qquad\square$

### B.4 Proof of Theorem 1

We first recall Theorem 1 below:

**Theorem 1.** *Let Assumptions 1, 2, 3 and 4 hold. Then, for any fixed $k$ with $k^* \leq k \leq d$, the I-OBS algorithm has the following local quadratic convergence rate:*

$$\|\theta_{t+1} - \theta^*\|_2 \leq \left(1 + \sqrt{\frac{L}{\mu}\frac{d-k^*}{d-k}}\right)\frac{M}{2\mu}\|\theta_t - \theta^*\|_2^2. \tag{9}$$

*Proof of Theorem 1.* From the update rule, it is easy to see that:

$$\theta_{t+1} - \theta^* = \theta_t - \mathbf{H}_t^{-1}\nabla f(\theta_t) - \theta^* - \mathbf{H}_t^{-1}\mathbf{H}_t^{S^{t+1}}\left(\theta_t - \mathbf{H}_t^{-1}\nabla f(\theta_t)\right)$$

For simplicity, we define $Q^* = \text{supp}(\theta^*), Q^t = \text{supp}(\theta_t)$, and recall that by definition, we have

$$
\begin{aligned}
S^{t+1} &= \underset{S\subseteq[d],|S|=d-k}{\arg\min} \ (\theta_t - \mathbf{H}_t^{-1}\nabla f(\theta_t))^\top \mathbf{H}_t^S(\theta_t - \mathbf{H}_t^{-1}\nabla f(\theta_t))\\
&= \underset{S\subseteq[d],|S|=d-k}{\arg\min} \ (\theta_t - \mathbf{H}_t^{-1}\nabla f(\theta_t))^T \mathbf{I}_S^\top(\mathbf{I}_S\mathbf{H}_t^{-1}\mathbf{I}_S^\top)^{-1}\mathbf{I}_S(\theta_t - \mathbf{H}_t^{-1}\nabla f(\theta_t)).
\end{aligned}
$$

Then we have

$$\|\theta_{t+1} - \theta^*\|_2 \leq \|\theta_t - \mathbf{H}_t^{-1}\nabla f(\theta_t) - \theta^*\|_2 + \|\mathbf{H}_t^{-1}\mathbf{H}_t^{S^{t+1}}\left(\theta_t - \mathbf{H}_t^{-1}\nabla f(\theta_t)\right)\|_2.$$

Next we want to control $\|\mathbf{H}_t^{-1}\mathbf{H}_t^{S^{t+1}}\left(\theta_t - \mathbf{H}_t^{-1}\nabla f(\theta_t)\right)\|_2$ term above. For simplicity, we define $\theta^+ := \theta_t - \mathbf{H}_t^{-1}\nabla f(\theta_t)$.

$$
\begin{aligned}
\left\|\mathbf{H}_t^{-1}\mathbf{H}_t^{S^{t+1}}\theta^+\right\|_2^2 &= (\theta^+)^\top\mathbf{H}_t^{S^{t+1}}\mathbf{H}_t^{-2}\mathbf{H}_t^{S^{t+1}}\theta^+\\
&= (\theta^+)^\top(\mathbf{H}_t^{S^{t+1}})^{1/2}(\mathbf{H}_t^{S^{t+1}})^{1/2}\mathbf{H}_t^{-2}(\mathbf{H}_t^{S^{t+1}})^{1/2}(\mathbf{H}_t^{S^{t+1}})^{1/2}\theta^+\\
&\leq \lambda_{\max}\left((\mathbf{H}_t^{S^{t+1}})^{1/2}\mathbf{H}_t^{-2}(\mathbf{H}_t^{S^{t+1}})^{1/2}\right)(\theta^+)^\top\mathbf{H}_t^{S^{t+1}}\theta^+\\
&= \lambda_{\max}\left(\mathbf{H}_t^{-2}\mathbf{H}_t^{S^{t+1}}\right)(\theta^+)^\top\mathbf{H}_t^{S^{t+1}}\theta^+\\
&= \lambda_{\max}\left(\mathbf{H}_t^{-1}\mathbf{H}_t^{S^{t+1}}\mathbf{H}_t^{-1}\right)(\theta^+)^\top\mathbf{H}_t^{S^{t+1}}\theta^+\\
&= \left\|\mathbf{H}_t^{-1}\mathbf{H}_t^{S^{t+1}}\mathbf{H}_t^{-1}\right\|_2(\theta^+)^\top\mathbf{H}_t^{S^{t+1}}\theta^+,
\end{aligned}
$$

where we used the fact that matrices $\mathbf{AB}$ and $\mathbf{BA}$ have the same nonzero eigenvalues counting multiplicity, and for any symmetric and positive semi-definite matrix $\mathbf{A}$ it holds $\|\mathbf{A}\|_2 = \lambda_{\max}(\mathbf{A})$. The norm $\|\mathbf{H}_t^{-1}\mathbf{H}_t^{S^{t+1}}\mathbf{H}_t^{-1}\|_2$ can be bounded by splitting it into two matrix operations by invoking Lemma 7 and strong convexity Assumption 2:

$$\left\|\mathbf{H}_t^{-1}\mathbf{H}_t^{S^{t+1}}\mathbf{H}_t^{-1}\right\|_2 \leq \left\|\mathbf{H}_t^{-1}\mathbf{H}_t^{S^{t+1}}\right\|_2\|\mathbf{H}_t^{-1}\|_2 \leq \frac{1}{\mu}.$$

Furthermore, based on (24) we have $(\theta^+)^\top\mathbf{H}_t^{S^{t+1}}\theta^+ \leq (\theta^+)^\top\mathbf{H}_t^{\tilde{S}}\theta^+$ for any set $\tilde{S}$ of the same cardinality as $S^{t+1}$. Noticing that $\mathbf{I}_{\tilde{S}}\mathbf{E}_{\tilde{S}} = \mathbf{I}_{\tilde{S}}$, we get

$$
\begin{aligned}
(\theta^+)^\top\mathbf{H}_t^{\tilde{S}}\theta^+ &= (\mathbf{E}_{\tilde{S}}\theta^+)^\top\mathbf{H}_t^{\tilde{S}}(\mathbf{E}_{\tilde{S}}\theta^+)\\
&= (\mathbf{E}_{\tilde{S}}\theta^+)^\top\mathbf{H}_t^{1/2}\mathbf{H}_t^{-1/2}\mathbf{H}_t^{\tilde{S}}\mathbf{H}_t^{-1/2}\mathbf{H}_t^{1/2}(\mathbf{E}_{\tilde{S}}\theta^+)\\
&\leq \lambda_{\max}\left(\mathbf{H}_t^{-1/2}\mathbf{H}_t^{\tilde{S}}\mathbf{H}_t^{-1/2}\right)(\mathbf{E}_{\tilde{S}}\theta^+)^\top\mathbf{H}_t(\mathbf{E}_{\tilde{S}}\theta^+)\\
&= \lambda_{\max}\left(\mathbf{H}_t^{-1}\mathbf{H}_t^{\tilde{S}}\right)(\mathbf{E}_{\tilde{S}}\theta^+)^\top\mathbf{H}_t(\mathbf{E}_{\tilde{S}}\theta^+)\\
&\leq L\left\|\mathbf{E}_{\tilde{S}}\theta^+\right\|_2^2 = L\left\|\mathbf{I}_{\tilde{S}}\theta^+\right\|_2^2,
\end{aligned}
$$

where we used the fact that $\mathbf{H}_t^{-1}\mathbf{H}_t^{\tilde{S}}$ is a projection matrix (Lemma 7) and restricted first-order smoothness condition (Assumption 3). Then, choosing $\tilde{S} = \arg\min_{S:|S|=d-k} \|\mathbf{I}_S\theta^+\|_2^2$ and using Lemma 8, we have that:

$$
\begin{aligned}
\left\|\mathbf{I}_{\tilde{S}}(\theta_t - \mathbf{H}_t^{-1}\nabla f(\theta_t))\right\|_2^2 &= \|T_k(\theta_t - \mathbf{H}_t^{-1}\nabla f(\theta_t)) - (\theta_t - \mathbf{H}_t^{-1}\nabla f(\theta_t))\|_2^2 \\
&\leq \frac{d - k^*}{d - k}\|\theta_t - \mathbf{H}_t^{-1}\nabla f(\theta_t) - \theta^*\|_2^2.
\end{aligned}
$$

Hence, combining these bounds we arrive

$$
\|\mathbf{H}_t^{-1}\mathbf{H}_t^{S^{t+1}}\left(\theta_t - \mathbf{H}_t^{-1}\nabla f(\theta_t)\right)\|_2 \leq \sqrt{\frac{L}{\mu}\frac{d - k^*}{d - k}}\|\theta_t - \mathbf{H}_t^{-1}\nabla f(\theta_t) - \theta^*\|_2.
$$

And the rest follows from the standard proof of Newton's methods, see e.g. [Nesterov, 2018, pg. 38-39]. We redo the proof here for completeness:

$$
\begin{aligned}
\|\theta_t - \mathbf{H}_t^{-1}\nabla f(\theta_t) - \theta^*\|_2 &\leq \left\|\theta_t - \theta^* - \mathbf{H}_t^{-1}\int_0^1 \nabla^2 f(\theta^* + \tau(\theta_t - \theta^*))(\theta_t - \theta^*)\, d\tau\right\|_2 \\
&= \left\|\mathbf{H}_t^{-1}\int_0^1 \left(\nabla^2 f(\theta_t) - \nabla^2 f(\theta^* + \tau(\theta_t - \theta^*))\right)(\theta_t - \theta^*)\, d\tau\right\|_2 \\
&\leq \|\mathbf{H}_t^{-1}\|_2 \left\|\int_0^1 \left(\nabla^2 f(\theta_t) - \nabla^2 f(\theta^* + \tau(\theta_t - \theta^*))\right)\, d\tau\right\|_2 \|\theta_t - \theta^*\|_2 \\
&\leq \frac{1}{\mu}\cdot M\|\theta_t - \theta^*\|_2 \int_0^1 \tau\, d\tau \cdot \|\theta_t - \theta^*\|_2 \\
&\leq \frac{M}{2\mu}\|\theta_t - \theta^*\|_2^2.
\end{aligned}
$$

Finally, combining this with the previous bounds we get

$$
\begin{aligned}
\|\theta_{t+1} - \theta^*\|_2 &\leq \|\theta_t - \mathbf{H}_t^{-1}\nabla f(\theta_t) - \theta^*\|_2 + \|\mathbf{H}_t^{-1}\mathbf{H}_t^{S^{t+1}}\left(\theta_t - \mathbf{H}_t^{-1}\nabla f(\theta_t)\right)\|_2 \\
&\leq \left(1 + \sqrt{\frac{L}{\mu}\frac{d - k^*}{d - k}}\right)\|\theta_t - \mathbf{H}_t^{-1}\nabla f(\theta_t) - \theta^*\|_2 \\
&\leq \left(1 + \sqrt{\frac{L}{\mu}\frac{d - k^*}{d - k}}\right)\cdot\frac{M}{2\mu}\|\theta_t - \theta^*\|_2^2,
\end{aligned}
$$

which concludes the proof of Theorem 1. $\qquad\square$

### B.5 Proof of Lemma 3

**Lemma 3.** *Let Assumptions 1, 2, 3 and 4 hold. Then I-OBS method defined in (12) has the following convergence rate:*

$$
\|\theta_{t+1} - \theta_*\|_2 \leq \left(1 + \sqrt{\frac{k^*}{k}}\right)\frac{M}{2\mu}\|\theta_t - \theta^*\|_2^2 \tag{13}
$$

*Proof of Lemma 3.* Similar to the proof of Theorem 1, we define $Q^t = \text{supp}(\theta_t)$, $Q^* = \text{supp}(\theta^*)$. Notice that by definition

$$
\theta_{t+1} = T_k(\theta_t - \mathbf{H}_t^{-1}\nabla f(\theta_t)) = \mathbf{E}_{Q^{t+1}}(\theta_t - \mathbf{H}_t^{-1}\nabla f(\theta_t)). \tag{25}
$$

Then we have that

$$
\begin{aligned}
\|\theta_{t+1} - \theta^*\|_2 &= \|\mathbf{E}_{Q^{t+1}\cup Q^*}(\theta_{t+1} - \theta^*)\|_2 \\
&= \|\mathbf{E}_{Q^{t+1}\cup Q^*}(T_k(\theta_t - \mathbf{H}_t^{-1}\nabla f(\theta_t)) - \theta^*)\|_2 \\
&\leq \|\mathbf{E}_{Q^{t+1}\cup Q^*}(T_k(\theta_t - \mathbf{H}_t^{-1}\nabla f(\theta_t)) - \mathbf{E}_{Q^{t+1}\cup Q^*}(\theta_t - \mathbf{H}_t^{-1}\nabla f(\theta_t))\|_2 \\
&\quad + \|\mathbf{E}_{Q^{t+1}\cup Q^*}(\theta_t - \mathbf{H}_t^{-1}\nabla f(\theta_t) - \theta^*)\|_2. \tag{26}
\end{aligned}
$$

We further upper bound the second term by dropping the projection matrix $\mathbf{E}_{Q^{t+1} \cup Q^*}$. For the first term, (25) implies

$$\mathbf{E}_{Q^{t+1} \cup Q^*} T_k(\theta_t - \mathbf{H}_t^{-1} \nabla f(\theta_t)) = T_k \left( \mathbf{E}_{Q^{t+1} \cup Q^*} (\theta_t - \mathbf{H}_t^{-1} \nabla f(\theta_t)) \right).$$

Hence, the first term can be upper bounded by Lemma 8:

$$\begin{aligned}
& \| \mathbf{E}_{Q^{t+1} \cup Q^*} (T_k(\theta_t - \mathbf{H}_t^{-1} \nabla f(\theta_t)) - \mathbf{E}_{Q^{t+1} \cup Q^*} (\theta_t - \mathbf{H}_t^{-1} \nabla f(\theta_t)) \|_2^2 \\
&= \| T_k \left( \mathbf{E}_{Q^{t+1} \cup Q^*} (\theta_t - \mathbf{H}_t^{-1} \nabla f(\theta_t)) \right) - \mathbf{E}_{Q^{t+1} \cup Q^*} (\theta_t - \mathbf{H}_t^{-1} \nabla f(\theta_t)) \|_2^2 \\
&\leq \frac{k^*}{k} \| \mathbf{E}_{Q^{t+1} \cup Q^*} (\theta_t - \mathbf{H}_t^{-1} \nabla f(\theta_t)) - \theta^* \|_2^2 \\
&\leq \frac{k^*}{k} \| \theta_t - \mathbf{H}_t^{-1} \nabla f(\theta_t) - \theta^* \|_2^2
\end{aligned}$$

Using the obtained bounds, (26) implies

$$\|\theta_{t+1} - \theta^*\|_2 \leq \left( 1 + \sqrt{\frac{k^*}{k}} \right) \|\theta_t - \mathbf{H}_t^{-1} \nabla f(\theta_t) - \theta^*\|_2.$$

The rest follows from standard analysis of Newton's methods showing that

$$\|\theta_t - \mathbf{H}_t^{-1} \nabla f(\theta_t) - \theta^*\|_2 \leq \frac{M}{2\mu} \|\theta_t - \theta^*\|_2^2.$$

$\square$

### B.6   Proof of Lemma 4

**Lemma 4.** *With fixed sparsity constraint $\mathbf{I}_S \theta = 0$, the stochastic quadratic problem*

$$\arg\min_{\theta : \mathbf{I}_S \theta = 0} \langle g(\theta_t), \theta - \theta_t \rangle + \frac{|\langle g(\theta_t), \theta - \theta_t \rangle|^2}{2\|g(\theta_t)\|_2} + \frac{\lambda}{2} \|\theta - \theta_t\|^2$$

*has the following solution for the update*

$$\theta_{t+1} = \mathbf{E}_Q \left( \theta_t - \frac{1}{\lambda + \|\mathbf{E}_Q g(\theta_t)\|_2^2 / \|g(\theta_t)\|_2} g(\theta_t) \right),$$

*where the mask $Q$ is the complement of $S$.*

*Proof of Lemma 4.*   Due to the convexity of the quadratic objective and linearity of constraints, we solve the problem with KKT conditions. The Lagrangian with multipliers $\xi \in \mathbb{R}^{|S|}$ will be

$$L(\theta, \xi) = g(\theta_t)^\top (\theta - \theta_t) + \frac{|g(\theta_t)^\top (\theta - \theta_t)|^2}{2\|g(\theta_t)\|_2} + \frac{\lambda}{2} \|\theta - \theta_t\|_2^2 + \xi^\top \mathbf{I}_S \theta \tag{27}$$

Taking the gradient with respect to each variable $\theta$ and $\xi$ separately, we have

$$0 = \nabla_\theta \mathcal{L}(\theta, \xi) = g(\theta_t) + \left( \frac{g(\theta_t) g(\theta_t)^\top}{\|g(\theta_t)\|_2} + \lambda \mathbf{I} \right) (\theta - \theta_t) + \mathbf{I}_S^\top \xi$$

$$0 = \nabla_\xi \mathcal{L}(\theta, \xi) = \mathbf{I}_S \theta$$

Multiplying the first equation by $\mathbf{I}_S$, we solve for $\xi$:

$$\xi = -\mathbf{I}_S \left( g(\theta_t) + \left( \frac{g(\theta_t) g(\theta_t)^\top}{\|g(\theta_t)\|_2} + \lambda \mathbf{I} \right) (\theta - \theta_t) \right)$$

Then we plug it back in the first equation

$$
\begin{aligned}
0 &= g(\theta_t) + \left( \frac{g(\theta_t)g(\theta_t)^\top}{\|g(\theta_t)\|_2} + \lambda \mathbf{I} \right)(\theta - \theta_t) - \mathbf{I}_S^\top \mathbf{I}_S \left( g(\theta_t) + \left( \frac{g(\theta_t)g(\theta_t)^\top}{\|g(\theta_t)\|_2} + \lambda \mathbf{I} \right)(\theta - \theta_t) \right) \\
&= (\mathbf{I} - \mathbf{E}_S) \left[ g(\theta_t) + \left( \frac{g(\theta_t)g(\theta_t)^\top}{\|g(\theta_t)\|_2} + \lambda \mathbf{I} \right)(\theta - \theta_t) \right] \\
&= \mathbf{E}_Q \left[ g(\theta_t) \left( 1 + \frac{g(\theta_t)^\top (\theta - \theta_t)}{\|g(\theta_t)\|_2} \right) + \lambda(\theta - \theta_t) \right].
\end{aligned}
\tag{28}
$$

From this and the fact that $\text{supp}(\theta) \subseteq Q$ (since $I_S \theta = 0$) implies that:

$$
\theta - \theta_t = \mathbf{E}_Q(\theta - \theta_t) = -\eta \mathbf{E}_Q g(\theta_t)
\tag{29}
$$

for some scalar value $\eta$. It remains to derive the expression for $\eta$, which can be done by plugging the obtained expression for $\theta - \theta_t = -\eta \mathbf{E}_Q g(\theta_t)$ into (28) and solving for the parameter $\eta$.

$$
\begin{aligned}
0 &= \mathbf{E}_Q \left[ g(\theta_t) \left( 1 + \frac{g(\theta_t)^\top (\theta - \theta_t)}{\|g(\theta_t)\|_2} \right) + \lambda(\theta - \theta_t) \right] \\
&= \mathbf{E}_Q g(\theta_t) \left( 1 + \frac{g(\theta_t)^\top (\theta - \theta_t)}{\|g(\theta_t)\|_2} \right) + \lambda \mathbf{E}_Q(\theta - \theta_t) \\
&= \mathbf{E}_Q g(\theta_t) \left( 1 - \eta \frac{g(\theta_t)^\top \mathbf{E}_Q g(\theta_t)}{\|g(\theta_t)\|_2} \right) - \eta \lambda \mathbf{E}_Q g(\theta_t) \\
&= \mathbf{E}_Q g(\theta_t) \left( 1 - \eta \frac{\|\mathbf{E}_Q g(\theta_t)\|_2^2}{\|g(\theta_t)\|_2} - \eta \lambda \right) \implies \eta = \frac{1}{\lambda + \|\mathbf{E}_Q g(\theta_t)\|_2^2 / \|g(\theta_t)\|_2}.
\end{aligned}
$$

The update rule (29) with the obtained expression of $\eta$ completes the proof. $\qquad\square$

## C  Technical Lemmas

**Lemma 5.** *Given a $\mu$-strongly convex and $L$-smooth function $f$, with the unique global minimizer $\theta^*$, we have the following properties:*

1. *$f$ satisfy $\mu$-PL inequality, namely for any $\theta$:*

$$
\|\nabla f(\theta)\|_2^2 \geq 2\mu(f(\theta) - f(\theta^*))
$$

2. *$f$ is 'almost' are quadratic function, for any $\theta$:*

$$
\frac{\mu}{2}\|\theta - \theta^*\|_2^2 \leq f(\theta) - f(\theta^*) \leq \frac{L}{2}\|\theta - \theta^*\|_2^2
$$

**Lemma 6.** *Given a twice differentiable function $f : \mathbb{R}^d \to \mathbb{R}$, assume $f$ satisfy assumptions 2, 3, then we have the following properties:*

1. *$\mathbf{H} := \nabla^2 f(\theta)$ exists and is positive definite for any $\theta$.*

2. *Given $S \in [d]$ with $|S| = k$ for any $0 < k \leq d$, given any $\alpha > 0$, then $\mathbf{H}_\alpha^S = I_S \mathbf{H}^{-\alpha} I_S^\top$ is positive definite, with:*

$$
L^{-\alpha} \leq \lambda_1(\mathbf{H}_\alpha^S) \leq \cdots \leq \lambda_k(\mathbf{H}_\alpha^S) \leq \mu^{-\alpha}
$$

*Proof.* The first argument is directly from the strong convexity assumption.

For the second argument, the upper bound directly follows from the strong convexity assumption 2 and Lemma 9. Indeed, we have:

$$
\lambda_k(\mathbf{H}_\alpha^S) \leq \lambda_d(\mathbf{H}^{-\alpha}) = \lambda_1(\mathbf{H})^{-\alpha} = \mu^{-\alpha}
$$

Next, we show that $\lambda_1(\mathbf{H}^{-\alpha}) \geq L^{-\alpha}$ under assumption 3. Denote $v_1, ..., v_d$ the eigenvectors corresponding to $\lambda_1(\mathbf{H}), ..., \lambda_d(\mathbf{H})$, and let $\|v_i\|_2 = 1, \forall i \in [d]$ w.l.o.g. Given any vector $u \in \mathbb{R}^k$ such that $\|u\|_2 = 1$, we have that:

$$u^\top \mathbf{H}_\alpha^S u = (I_S^\top u)^\top \mathbf{H}^{-\alpha}(I_S^\top u) = \sum_{i=1}^d \lambda_i(\mathbf{H})^{-\alpha}((I_S u)^\top v_i)^2$$

Note that $\sum_{i=1}^d ((I_S^\top u)^\top v_i)^2 = \|u\|_2^2 = 1$, and the function $g(x) = x^{-\alpha}$ is convex for $x > 0$. Thus we have:

$$
\begin{aligned}
u^\top \mathbf{H}_\alpha^S u &= \sum_{i=1}^d \lambda_i(\mathbf{H})^{-\alpha}((I_S^\top u)^\top v_i)^2 \\
&\geq (\sum_{i=1}^d \lambda_i(\mathbf{H})((I_S^\top u)^\top v_i)^2)^{-\alpha} \\
&= ((I_S^\top u)^\top \mathbf{H}(I_S^\top u))^{-\alpha} \geq L^{-\alpha}
\end{aligned}
$$

where the first inequality is due to Jensen's inequality, and the last inequality follows from assumption 3. The lower bound also implies that $H_\alpha^S$ is positive definite.

$\square$

**Lemma 7.** *For any set $S$, we have $\|\mathbf{H}_t^{-1}\mathbf{H}_t^S\|_2 \leq 1$.*

*Proof.* Using the definition of $\mathbf{H}_t^S$, we can see that the matrix $P = \mathbf{H}_t^{-1}\mathbf{H}_t^S$ is a projection matrix (i.e., $P^2 = P$) with eigenvalues 0 or 1. Hence, $\|Px\| \leq \|x\|$ for any vector $x \in \mathbb{R}^d$. $\square$

**Lemma 8** (Property of top-$k$ operator). *[Peste et al., 2021, Lemma 1] Let $u, v$ be vectors with sparsity parameters $k_u$ and $k_v$ respectively such that $k_v < k_u$. Then, for any $k \in [k_v, k_u]$ we have:*

$$\|T_k(u) - u\|_2^2 \leq \frac{k_u - k}{k_u - k_v}\|u - v\|_2^2.$$

*Proof of Lemma 8.* We reprove the lemma here for completeness. The case $k = k_u$ trivially holds, so let us assume that $k < k_u$. Notice that the ratio

$$\frac{\|T_k(u) - u\|_2^2}{k_u - k}$$

representing the average of squared coordinates is monotonically decreasing as we increase $k$. Hence,

$$\frac{\|T_k(u) - u\|_2^2}{k_u - k} \leq \frac{\|T_{k_v}(u) - u\|_2^2}{k_u - k_v}. \tag{30}$$

It remains to notice that, $T_{k_v}(u)$ is the closest $k_v$-sparse vector to $u$ with respect to $l_2$ norm, namely $\|T_{k_v}(u) - u\|_2 \leq \|v - u\|_2$. Plugging this inequality in (30) we conclude the lemma. $\square$

**Lemma 9** (Interlacing property). *[Horn and Johnson, 2012, Theorem 4.3.28 restated] Consider a symmetric matrix $\mathbf{H} \in \mathbb{R}^{d \times d}$, with eigenvalues $\lambda_1(\mathbf{H}) \leq ... \leq \lambda_d(\mathbf{H})$. Given $S \subseteq [d]$, we have the following properties on eigenvalue of matrix $\mathbf{I}_S \mathbf{H} \mathbf{I}_S^\top \in \mathbb{R}^{|S| \times |S|}$ :*

$$\lambda_i(\mathbf{H}) \leq \lambda_i(\mathbf{I}_S \mathbf{H} \mathbf{I}_S^\top) \leq \lambda_{i+d-|S|}(\mathbf{H}), \quad \forall i = 1, ..., |S|$$

