# OpenReview forum: "The Iterative Optimal Brain Surgeon: Faster Sparse Recovery by Leveraging Second-Order Information"
_NeurIPS.cc/2024/Conference — NeurIPS 2024 poster_

### Official Review · Reviewer_3y7g · 2024-07-10

**Soundness:** 3
**Presentation:** 4
**Contribution:** 3
**Rating:** 7
**Confidence:** 4

**Summary:**

This paper proposes a theoretically convergent iterative Optimal Brain Surgeon (OBS) algorithm, which generalizes the classic Iterative Hard Thresholding (IHT)-based algorithms by incorporating approximate second-order information in the sparse projection step. The author also provides practical variants of these algorithms for solving sparse linear regression and model pruning. The experiments show that the proposed algorithm leads to faster convergence than traditional algorithms and improves accuracy.

**Strengths:**

The proposed algorithm is simple yet effective.

**Weaknesses:**

Theoretical analysis does not hold in general for practical problems.

**Questions:**

Detailed comments are as follows:
1. The author elaborates on the theoretically convergent I-OBS method and provides specific calculations for both practical and theoretical schemes. What is the gap between the two schemes? Can it be verified through experiments or illustrated with examples? What is the specific operator T in the practical scheme?
2. In lines 158 and 169, the specific form of the model \( \phi_t(\theta) \) differs.
3. In the model pruning experiment, the author provides the specific algorithm Alg. 2 for the practical use of I-OBS. By constructing a constrained optimization problem for the learnable parameters \( W \) of each layer to fine-tune the model, the author has verified the method's feasibility through experiments. Is it theoretically feasible?

**Limitations:**

The author objectively analyzed the limitations of this article.

---

> ### Author Rebuttal · Authors · 2024-08-06
>
> Thank you for the valuable questions and positive comments. We address your questions below:
>
> **Q1, Regarding the gap between the practical and theoretical schemes:**
>
> We note that for the theoretical scheme we have $|| \theta\_{t+1} - \theta^* ||\_2 \leq \left(1+ \sqrt{\frac{L}{\mu}\frac{d-k^*}{d-k}} \right) \frac{M}{2 \mu} || \theta\_t - \theta^*||\_2^2.$ (Theorem 1) and for the practical scheme we have $ || \theta\_{t+1} - \theta\_* ||\_2 \leq \left( 1+ \sqrt{\frac{k^*}{k}} \right) \frac{M}{2 \mu} || \theta\_t - \theta^* ||\_2^2$ ( Lemma 3). In both cases, the upper is in the form of $|| \theta_{t+1} - \theta_* ||_2 \leq C|| \theta_t - \theta^* ||_2^2 $, where $C$ is a constant. Thus, both scheme gives a super-exponential local convergence rate, namely both algorithms achieve $\epsilon$-error with initialization $||\theta_0 - \theta^*||_2 \leq \frac{1}{2 C}$ in $O( \log \log \frac{1}{\epsilon})$ steps. Thus,  there is no gap in terms of convergence rate. There is indeed a difference in the convergence radius $ \frac{1}{2 C}$ since the constant $C$ is different, but we remark that this difference is minor in practice.
>
> **Q2, In lines 158 and 169, the specific form of the model ( $\phi_t (\theta)$ ) differs**
>
> We thank the reviewer for the detailed review, however, there are no $ \phi_t(\theta)$ in line 169. Do you mean the $ \phi_t(\theta)$ in line 164?
> In line 164, we replace the gradient  $\nabla f(\theta_t) $ term with the stochastic gradient term$g_t(\theta_t) $, so that if one solves the proximal problem induced by the $ \phi_t(\theta)$ in line 164, one will get the update of stochastic gradient descent rather than gradient descent.
> Please let us know if there are any further questions regarding this point.
>
> **Q3, Regarding the theoretical analysis of the practical implementation of I-OBS**
>
> While we believe it would be interesting to obtain a theoretical guarantee for the practical implementation of I-OBS, we find it is difficult to make it end-to-end rigorous, as the underlying constrained optimization problem is NP-hard. In particular, to efficiently solve the constraint optimization problem in line 7 of Algorithm 2, we applied existing heuristic quadratic sparsity solvers, such as OBC or SparseGPT. (In preliminary experiments, we have also investigated applying algorithms with approximation guarantees, such as Natarajan’s algorithm [Natarajan95], but found that these are much less scalable and do not provide better solutions.) These solvers use greedy heuristics to efficiently compute the masks, since obtaining the theoretical optimal masks shown in line 7 of Algorithm 1 is practically infeasible. Despite this approximation, we note that our approach obtains high practical performance.
>
> [Natarajan95] Natarajan, Balas Kausik. "Sparse approximate solutions to linear systems." SIAM journal on computing 24.2 (1995): 227-234.

---

### Official Review · Reviewer_T9TW · 2024-07-11

**Soundness:** 3
**Presentation:** 4
**Contribution:** 2
**Rating:** 6
**Confidence:** 3

**Summary:**

This work combines second-order curvature information with sparse recovery algorithms to demonstrate, both theoretically and empirically, that the curvature information leads to improved convergence rates and generalization performance in post-training iterative pruning and sparse recovery

**Strengths:**

* This work offers a rigorous analysis of the proposed method, I-OBS, complete with detailed derivations and proofs.
* The authors ground their contributions within the context of the existing literature but clearly identify how it differs from prior works by leveraging second order information.
* Despite the intractability of the theoretical approach, the authors offer a practical formulation that yields improved performance across a diverse range of data domains.
* The paper is rather dense; however, the authors provide all the necessary notation required to follow their derivations.
* This is a well motivated line of inquiry as post-training compression is increasingly important in the era of LLMs. Further, despite requiring additional iterations compared to one-shot pruning methods, I-OBS converges in a small number of iterations (<100 for DeiT and <3 for LLMs).

**Weaknesses:**

* Grounding this work in some practical considerations would improve the overall impact of the paper and may help the reader assess whether this technique is suitable for a desired use case. For instance, adding some overall runtime characteristics for the empirical results would establish an order-of-magnitude estimate of the overall computational requirements.
* In a similar vein, conducting experiments with fine-grained sparsity such as 2:4 would be a nice extension to produce sparse neural networks that can actually be accelerated in practice. As it currently stands, the unstructured networks learned by I-OBS do not offer much in the way of an immediate practical application.
* For the LLM experiments, perplexity has been shown to be a somewhat misleading metric when evaluating compressed LLMs [1]. Ideally, the LLM experiments should be evaluated on downstream tasks such as GLUE or better yet the LLM-KICK benchmark [1].
* Several post-training compression schemes have been proposed recently. Comparisons with methods such as [2-6] would improve my confidence in the significance of this work.
* Several small typos, see suggestions below.

[1] A. Jaiswal, Z. Gan, X. Du, B. Zhang, Z. Wang, and Y. Yang, “Compressing LLMs: The Truth is Rarely Pure and Never Simple.” arXiv, Oct. 02, 2023. doi: 10.48550/arXiv.2310.01382.

[2] T. Dettmers et al., “SpQR: A Sparse-Quantized Representation for Near-Lossless LLM Weight Compression.” arXiv, Jun. 05, 2023. [Online]. Available: http://arxiv.org/abs/2306.03078

[3] M. Sun, Z. Liu, A. Bair, and J. Z. Kolter, “A Simple and Effective Pruning Approach for Large Language Models.” arXiv, Jun. 20, 2023. doi: 10.48550/arXiv.2306.11695.

[4] Y. Zhang, H. Bai, H. Lin, J. Zhao, L. Hou, and C. V. Cannistraci, “Plug-and-Play: An Efficient Post-training Pruning Method for Large Language Models,” presented at the The Twelfth International Conference on Learning Representations, Oct. 2023. [Online]. Available: https://openreview.net/forum?id=Tr0lPx9woF

[5] Y. Zhang et al., “Dynamic Sparse No Training: Training-Free Fine-tuning for Sparse LLMs.” arXiv, Oct. 17, 2023. doi: 10.48550/arXiv.2310.08915.

[6] Y. Ma et al., “AffineQuant: Affine Transformation Quantization for Large Language Models.” arXiv, Mar. 19, 2024. doi: 10.48550/arXiv.2403.12544.

**Questions:**

## Questions
* In terms of wall clock time, what was the duration required for the pruning results in Tables 1 and 2 and what hardware configuration was used?
* How do the compressed LLMs compare on downstream tasks such as LLM-KICK or GLUE?
* Can I-OBS be extended to fine-grained sparsity types such as N:M sparsity?
* Are the results in Table 2 obtained with 32- or 16-bit weights? How does a quantized dense model with half the precision as used in Table 2 compare?

## Suggestions
* Use vector graphic formats for Figure 1.
* Potential typos to address:
	* L281 & Figure 1 Caption: Refers to topk-WoodFisher, I believe this should be **topk-I-OBS**? Perhaps a prior naming convention?
	* Table 2 caption: Phi-1.5M -> Phi-1.5B
	* L332: pwer-layer -> per-layer
	* L340: mode -> model

**Limitations:**

In general the limitations listed are appropriate. However, I encourage the authors to add some discussion about the practical requirements for their algorithm in terms of time and compute.

---

> ### Author Rebuttal · Authors · 2024-08-06
>
> Thank you  for the detailed review, as well as your valuable questions and comments. We address your questions and concerns below
>
> **Q1 and W1 Regarding the training time of the pruning results**
>
> For the experiments on ViTs (Table 1 in the paper), it took about 4 hours to run 100 iterations and the hardware we used was Nvidia A100 GPU with 80GB memory. For LLMs experiments,  we ran I-OBS on Llama-2 (7B) and Llama-3 (8B) models, and it takes around 1.5 hours per iteration (15h for 10 iterations). Unfortunately, we haven't recorded the time of running the experiments on OPT-125M and Phi-1.5 (Table 2 in the paper).
>
> **Q2 and W3 Evaluating the performance on LLM-KICK or GLUE**
>
> To address your concern and study the scalability of I-OBS, we ran I-OBS on Llama-2 (7B) and Llama-3 (8B) models. Specifically, we apply SparseGPT (50% sparsity) and finetune on the same calibration set used for Hessian estimates. We evaluate the performance on 5-shot MMLU, one of the tasks from LLM-KICK benchmark–we did this due to time constraints but will expand to the full benchmark for the next revision.
>
> Experimental results are provided in Table 1 and Table 2 of the PDF file we attached in the global rebuttal.
>
> One can observe that the quality of sparse solution significantly improves after first finetuning iteration, thus validating our approach. Moreover, there is small improvement during the next few iterations, followed by gradual performance deterioration afterward, which we explain due to overfitting on the calibration set. Specifically, I-OBS improves accuracy on MMLU by >1 point, a 20% reduction in error relative to the original dense model.
>
> **W4 Comparison to other  works in pruning**
>
> Thank you for the suggestion. As also suggested by reviewer JoZw, we provide a more detailed literature review. Please refer the answer to **Q2 of reviewer JoZw**. We apologize that we can not copy the whole response here due to the character limit.
>
> Also, as suggested by reviewer JoZw, we conducted experiments on  MobileNetV1 to compare to the CBS methods.  The results are shown in  Table 3 of the attached file in global rebuttal. Our experiments imply that, starting with 60\% sparsity, our I-OBS pruner outperforms CBS method on MobileNetV1 by a large margin: 2\% for 60\% sparsity, 5\% for 70\% sparsity and 12\% for 80\% sparsity. For low sparsities (30\% to 50\%), the two methods are comparable, since the accuracy difference is less than 0.5\%
>
> **Q3 and W1  Extend to fine-grained sparsity type**
>
> The practical implementation of I-OBS indeed extends to N:M sparsity. In particular, we apply OBC or SparseGPT as sparsity solvers for the layerwise pruning problem, and  OBC or SparseGPT applies to N:M sparsity. However, for the theoretical results, we are unable to extend the analysis to the masks selected based on N:M sparsity, we will leave this to future work.
>
>
>
> **Q4 Are the results in Table 2 obtained with 32- or 16-bit weights? How does a quantized dense model with half the precision as used in Table 2 compare?**
>
> The results in Table 2 are obtained with bfloat16 (half precision) weights, which is standard. Generally, we found the weight precision to be orthogonal to our results, as sparsification can be applied independently of the baseline weight representation, and always yields similar speedups (the maximum speedup due to 2:4 sparsity is 2x for INT8 and FP16).
> More broadly, we believe our iterative approach should be generalizable to compression via quantization as well: specifically, we could modify the projection step to perform quantization rather than sparsification.
>
> **Suggestions**
>
> Thank you for the suggestions. We will change the figure format and correct the typos in the revision
>
> **References**
> [CBS] Yu, X., Serra, T., Ramalingam, S., & Zhe, S.. “The combinatorial brain surgeon: pruning weights that cancel one another in neural networks.”  ICML 2022

---

> > ### Comment · Reviewer_T9TW · 2024-08-12
> >
> > Thank you for the rebuttal and additional experiments, I believe they will improve the manuscript. I have elected to maintain my original score.

---

### Official Review · Reviewer_Dt8W · 2024-07-12

**Soundness:** 3
**Presentation:** 3
**Contribution:** 3
**Rating:** 6
**Confidence:** 2

**Summary:**

The paper presents a new family of algorithms called Iterative Optimal Brain Surgeon (I-OBS), extending the post-training Optimal Brain Surgeon (OBS) framework to an iterative setting commonly used in sparse recovery. I-OBS algorithms utilize second-order information during the sparse projection step, enhancing convergence guarantees compared to classic algorithms like Iterative Hard Thresholding (IHT).

Contributions of the paper:

* Introduction of I-OBS, which improves classic IHT-based algorithms by incorporating approximate second-order information in the sparse projection step.

* Provision of faster analytical convergence rates for I-OBS under standard first- and second-order smoothness and strong convexity assumptions. It also offers theoretical guarantees for existing practical pruning algorithms such as WoodFisher and OBC.

* Development of practical versions of the I-OBS algorithms that relax theoretical constraints, making them easy to implement and scalable for large problem instances, such as compressing vision and language models.

**Strengths:**

* The paper introduces an algorithm with theoretical guarantees that its predecessors do not provide.

* It also offers a practical computational algorithm that approximates the theoretical one.

* The paper shows the benefit of the proposed algorithm in training sparse linear regression.

* The paper shows empirically that the proposed algorithm is applicable to prune large models in experiments and obtain promising results.

**Weaknesses:**

* The paper does not analyze the differences between the practical version of the algorithm and the theoretical one.

* Since the algorithm uses second-order information, it may have time complexity issues.

* The paper only compares the proposed method with SparseGPT. Hence, other existing methods for pruning can be also added.

**Questions:**

* How does the algorithm using second-order information compare with one using first-order information?

* Can you provide experiments comparing training time in the pruning process among the proposed method and existing works?

* How does the proposed algorithm apply to the Convolution network?

**Limitations:**

The authors adequately addressed the limitations.

---

> ### Author Rebuttal · Authors · 2024-08-06
>
> Thank you for the valuable questions and comments, we address the questions and comments below:
>
> **W1. The difference between the practical and theoretical version of the algorithm**
> We compare the practical and theoretical version of Algorithm 1 below:
> (1) The way of choosing the mask is different: For the theoretical version of the algorithm, we choose the mask in an optimal way, which is equivalent to solving the integer programming problem in Line 7 of Algorithm 1; for the practical version, we simply use the top-k mask to replace the optimal mask
> (2) The complexity of the two versions is different. For the theoretical one, solving the integer programming problem in Line 7 of Algorithm 1 is NP-hard; while for the practical version, computing a top-k mask only has $O(d)$ complexity
> (3) The local convergence rate of the two algorithms is the same: in both case we obtain a local convergence in the form of $||\theta\_{t+1} - \theta^*||\_2 \leq C||\theta\_t - \theta^*||\_2^2$ implies a $\mathcal{O}(\log \log(1/\epsilon))$ iteration complexity for achieving an $\epsilon$-error with initialization $||\theta\_0 - \theta^*||\_2 \leq \frac{1}{2C}$.
>
> **W2. The complexity issue by using second-order information**
> While second-order information is historically hard to apply at the scale of deep networks, notice that there have recently been several efficient variants that work in linear time and space (in the model dimension) and can therefore be scaled. Specifically, we apply the recent scalable M-FAC method [M-FAC] to approximate the inverse of the Hessian.
>
> **W3. Comparing I-OBS with other existing methods**
>
>  We provide extra experiments that compare our methods with [CBS].  In Table 3 of the attached file in global rebuttal, we provide results for I-OBS applied to MobileNetV1 model used in STR. Our experiment results implies that: starting with 60\% sparsity, our I-OBS pruner outperforms CBS method on MobileNetV1 by a large margin: 2\% for 60\% sparsity, 5\% for 70\% sparsity and 12\% for 80\% sparsity. For low sparsities (30\% to 50\%), the two methods are comparable, since the accuracy difference is less than 0.5\%
>
>
> **Q1. How does the algorithm using second-order information compare with one using first-order information?**
>
> In I-OBS, we solve a sparse optimization problem at each step with the objective function being the second-order approximation of the loss function, in particular, we have $\theta\_{t+1} = \arg \min_{\theta: ||\theta||\_0 \le k} \; \phi\_t(\theta) + \tfrac{1}{2}||\theta-\theta\_t||^2_{H_t}$. The Hessian of the objective funcion appears in the $\tfrac{1}{2}||\theta-\theta\_t||^2\_{H\_t}$ term. While in the first-order method such as k-IHt studied in [AC/DC], the algorithm solves a sparse optimization problem at each step with the objective function being the first-order approximation of the loss function, in particular $\theta\_{t+1} = \arg \min\_{\theta: ||\theta||\_0\le k} \; \phi\_t(\theta) + \tfrac{1}{2\eta}||\theta-\theta\_t||^2$, the Hessian is replaced by a quadratic term.
>
> Regarding the convergence rate, typical first-order methods such as k-IHt studied in [AC/DC] have achieved $\epsilon$-error in $O(\log \frac{1}{\epsilon})$ iterations for any initialization. The I-OBS proposed in this paper, achieved $\epsilon$-error in $O(\log\log \frac{1}{\epsilon})$ iterations with initialization $||\theta\_0 - \theta^*||\_2 \leq \frac{1}{2 C}$. In short, I-OBS have a faster local convergence rate but we are unable to provide a global convergence rate. The difference resembles the one between gradient descent and Newton's methods.
>
> **Q2 Comparing training time in the pruning process among the proposed method and existing works**
>
> Thank you for the suggestion. However, it is a bit difficult to directly compare the pruning time since it depends on the implementation of the algorithms and the device. We need to adapt the implementation of methods in other existing works on our device, which is a bit time-consuming. We report the pruning time of our methods below:
>
> For the experiments on ViTs (Table 1 in the paper), it took about 4 hours for running 100 iterations and the hardware we used was Nvidia A100 GPU with 80GB memory. For LLMs experiments, as suggested by reviewer T9TW,  we ran I-OBS on Llama-2 (7B) and Llama-3 (8B) models, and it takes around 1.5 hours per iteration (15h for 10 iterations). Unfortunately, we haven't recorded the time of running the experiments on OPT-125M and Phi-1.5 models (Table 2 in the paper).
>
> **Q3 Whether the proposed algorithm applies to CNNs**
>
> The proposed methods indeed apply to CNNs. In Algorithm 2, we use OBC as the quadratic sparse solver for the problem defined in line 7 of Algorithm 2, and OBC applies to convolution layers. In fact, the experiments we are doing to baseline our methods with CBS are on MobileNet_v1, and we find I-OBS improves the performance of CBS on pruning MobileNet_v1.
>
>
> **References**
>
> [M-FAC] Frantar, Elias, Eldar Kurtic, and Dan Alistarh. "M-fac: Efficient matrix-free approximations of second-order information." Neurips 2021.
>
> [AC\DC] Peste, Alexandra, et al. "Ac/dc: Alternating compressed/decompressed training of deep neural networks."  Neurips 2021
>
> [CBS] Yu, X., Serra, T., Ramalingam, S., & Zhe, S.. “The combinatorial brain surgeon: pruning weights that cancel one another in neural networks.”  ICML 2022

---

> > ### Comment · Reviewer_Dt8W · 2024-08-09
> > **Response to authors**
> >
> > Thank you for your detailed response.
> >
> > I will keep my initial rating due to the lack of domain knowledge. I believe Iterative Optimal Brain Surgeon (I-OBS) is a novel contribution and can be widely used.
> >
> > Best regards,

---

### Official Review · Reviewer_JoZw · 2024-07-12

**Soundness:** 3
**Presentation:** 2
**Contribution:** 2
**Rating:** 7
**Confidence:** 4

**Summary:**

Having clarified my concerns in their rebuttal, I have updated my score for acceptance.
----
This paper presents a variant of the classic Optimal Brain Surgeon (OBS) method to iteratively prune multiple weights of a neural network at once, with each step consisting of obtaining a pruning mask for the fraction of weights to be removed and then updating the remaining weights. The update step is framed as a sparse recovery problem to be solved using second-order methods based on the Hessian of the loss function. Two approaches are proposed for the selection of weights to prune as well as for the update of the remaining weights, with one having stronger guarantees and the other being more manageable in practice.

**Strengths:**

In my opinion, the core contribution here is the idea of iteratively changing the pruning mask and updating the remaining weights, which implies that the decision about which weights to prune is revisited at every step. This is a subtle departure from how this is typically done with post-training pruning (i.e., prune once, update once; or prune a smaller amount, update, and then repeat by pruning more).

Moreover, I appreciate that the authors discuss the two steps by providing methods that either have better guarantees (Option 1) or that are more tractable in practice (Option 2).

The connection with sparse recovery is also very interesting, although I wished it was made more explicit. After one paragraph in Section 2, this connection is leveraged for their algorithm but the authors do not provide greater insight on the algorithms used in sparse recovering.

**Weaknesses:**

The authors imply that their work is the first generalization of OBS for simultaneously pruning multiple weights. However, this was already explored before in an ICML 2022 paper as the Combinatorial Brain Surgeon (CBS) [1]. In that paper, the method is also broken down in two steps, the first step selecting which weights to prune (CBS-S), which is equivalent to selecting the support mask in I-OBS, and the second step determining how to update the remaining weights (CBS-U), which is equivalent to optimizing the parameters in I-OBS. Because of the intractability of the problem, CBS-S is also approached in a greedy way, similar to Top$k$ in I-OBS. The main difference would be that CBS-S uses a semi-greedy heuristic to vary the selection of weights, whereas I-OBS updates the weights after fixing a mask and consequently affects which weights would be chosen by the greedy selection in the next iteration.

[1] https://arxiv.org/abs/2203.04466

More generally, the authors only cite fours papers about network pruning in the last decade in their literature review, one of those being a survey and another two being from a same group. For the rest of the paper, they just refer back to these two papers from the same group as the state-of-the-art and do not benchmark with anything else. There has been several papers focused on OBS in the last decade, both before and following [1]. Moreover, there are other pruning methodologies with theoretical guarantees, such as those based on coresets.

The connection with sparse recovery is quite interesting, but it generalizes the algorithm much more than it would be need for OBS. Because the objective function is quadratic, my understanding is that the weights has a closed-form solution (see discussion in the WoodFisher paper, in [1] regarding CBS-U, and also in lines 211-212 of I-OBS).

Finally, the authors only present results for sparsity 50%, which is not a lot of pruning in practice. For that amount of pruning, magnitude pruning (preserving the half of the weights with largest absolute value) is competitive with more refined techniques, such as WoodFisher and CBS. Because the weight selection done by the authors with Top$k$ is basically magnitude pruning, the results do not provide evidence of the strength of the technique. The technique may be strong, but that needs to be shown in more detail.

Other minor comments about the writing:

6-7: "lack a solid theoretical understanding": this is a bold statement, perhaps a bit unfair, and I don't think that the theory provided in the paper addresses exactly that (you just prove convergence rates for an update based on an optimization algorithm)

Equations 7 and 8: What is $H_t$ used as a subindex?

201: "critierion" -> criterion

201: "[,] which recovers"

210: $X$ not in bold as in previous uses

215: "in the first step one": remove one?

216 and 217: it is not "bruteforcing" if what you are doing takes linear time; bruteforcing is solving an entire problem by exhaustion

222: "As observed Algorithm 1": please rewrite

251, 278, 304-305, 311, 346: these references should have been in parantheses rather than having the authors names mentioned in the sentence without proper wording for it

256: "call[ed]"

269: "the[o]retical"

282: remove comma before "due"

285: remove "to"

Algorithm 2, line 1: remove first "for each layer" and remove comma

339: "guarentee" -> guarantee

340: "mode[l] pruning"?

341: "those assumption": either use "this" or use "assumptions"

342: "is" -> to be

**Questions:**

1) Can you please address the overlap with CBS?

2) Can you please frame your contribution over a broader range of recent work on network pruning, besides the four papers that you cited?

3) Can you please discuss the connection with sparse recovery in more detail, and also the relevance of this connection if having a quadratic objective function as in I-OBS leads to an optimization problem that can solved in closed-form?

4) Can you please benchmark your approach with at least magnitude pruning and CBS?

5) If possible, can you please provide results with higher amounts of sparsity?

**Limitations:**

The authors did a reasonable work discussing limitations and what should be done in future work.

---

> ### Author Rebuttal · Authors · 2024-08-06
>
> Thank you for your detailed reviews, as well as the insightful comments and questions. We address your comments and questions below
>
> **Q1**. CBS assumes the model gradient is zero for post-training pruning, while I-OBS does not, making I-OBS more general. We retain the gradient term because models may not be fully trained and subsequent I-OBS iterations can increase gradient norms for sparsity. Additionally, I-OBS is an iterative algorithm, in contrast to CBS's one-shot approach, allowing CBS-S to be used as a projection step in our iterative process by taking the gradient term into consideration.
>
> The objectives for selecting the masks are also different. Specifically, in the IQP formulation of CBS-S procedure for mask selection, the remaining weights and their further updates are ignored. This means that the updates of the unpruned weights are not considered during the mask selection procedure. Consequently, solving each subproblem (CBS-S and CBS-U) to optimality does not guarantee optimal solution to CBS.
>
> In the theoretical I-OBS algorithm, masks and weight updates are solved jointly, as stated in Lemma 2. Optimization in equation (8) handles both new weights and masks. Although the practical I-OBS algorithm first selects a mask and then updates weights (similar to CBS-U), it optimally finds the mask by solving an Integer Programming (IP) problem, unlike CBS-S's IQP. This makes I-OBS mask selection optimal but computationally challenging, so the practical version uses a top-k mask for efficiency, maintaining the same local convergence rate in strongly convex cases.
>
> **Q2.** We will discuss more literature review over broader range of model pruning:
> Our work follows the salience-based weight pruning approach, which evaluates the impact of removing weights on the model's loss or output. Among these, methods based on second-order information are most relevant, particularly those following the [OBS] framework. [L-OBS] uses a second-order approximation of the loss function, assuming a zero gradient, and introduces a layerwise strategy to approximate the Hessian. [WoodFisher] scales this idea with the empirical Fisher approximation of the Hessian, improving performance and considering non-zero gradients. However, WoodTaylor methods in [WoodFisher] focus on pruning one weight at a time, whereas our work extends this to multiple weights.
>
> Other methods addressing multiple weight pruning include [CBS], which formulates mask selection as an integer programming problem and proposes two heuristics to approximate its solution. [OBC] tackles layerwise pruning with a quadratic problem and introduces a greedy heuristic for efficient layerwise problem-solving. Similarly, [CHITA] formulates layerwise pruning as an $L0L2$-regularized quadratic problem and proposes an IHT-like iterative algorithm with a line search strategy. [Net-trim] minimizes the $L1$​ norm of weight matrices while ensuring similar activations in the pruned layer. While prior iterative algorithms like [CHITA] focus on specific problems, our methods apply to general loss functions beyond quadratic problems and offer convergence guarantees. Practically, within our iterative framework, using a cost-effective projection step (e.g., TopK or SparseGPT) yields competitive results compared to more complex one-shot solvers.
>
> **Q3.** We consider sparse optimization problems, aiming to find the sparse solution $\theta_*$ for an objective function $f(\theta)$ with a $k^*$-sparse global optimum $\theta_*$. This includes the classic sparse recovery problem: recovering a $k$-sparse signal $\theta_*$ from a noisy observation $A \theta_* + \epsilon$ with a sensing matrix $A$.
> I-OBS is a sparse optimization algorithm for sparse recovery. Besides finding $\theta_*$, it ensures each iteration's weights are $k$-sparse (with $k \ge k^*$), guaranteeing a $k$-sparse solution even if $\theta_*$ is not found. Such type of algorithms are known as proper learners in sparse recovery. Both theoretical and practical versions of I-OBS are sparse optimization algorithms resembling sparse Newton's methods. Similar first-order algorithms are studied in [AC/DC], but I-OBS leverages second-order information to potentially speed up convergence (both I-OBS versions achieve super-exponential local convergence rates).
>
> For quadratic objectives, consider two settings: strongly convex (Hessian is positive definite) and convex (Hessian is positive semi-definite but not positive definite).
> Strongly convex: The theoretical I-OBS solves the problem in one step as the second-order approximation equals the function itself. The challenge is efficiently computing the optimal mask. The practical I-OBS doesn't converge in one step but shows super-exponential local convergence.
> Convex: I-OBS cannot be directly applied due to the non-invertible Hessian, similar to issues in Newton's method for convex optimization. Adding a small L2 regularization can address invertibility but complicates obtaining a convergence rate, as the global optimum is no longer sparse. This is left for future work.
>
> **Q4 and Q5.**  Thank you for the suggestion.  In Table 3 of the attached file in global rebuttal, we provide results for I-OBS applied to MobileNetV1 model used in STR, which is the same model studied in Table 4 of the CBS paper. We skip the depth-wise convolutions having shape $(C, 1, K, K)$ when we apply our pruning algorithm, as is standard. Starting with 60\% sparsity, our I-OBS pruner outperforms CBS method on MobileNetV1 by a large margin: 2\% for 60\% sparsity, 5\% for 70\% sparsity and 12\% for 80\% sparsity. For low sparsities (30\% to 50\%), the two methods are comparable, since the accuracy difference is less than 0.5\%.

---

> > ### Author Response · Authors · 2024-08-12
> > **Discussion gentle reminder**
> >
> > Dear reviewer,
> >
> > As the discussion period will end soon, we wanted to respectfully ask if you could please provide feedback on our responses to your concerns, and specifically on the additional experimental results provided.
> >
> > Best regards,\
> > The authors

---

> > ### Comment · Reviewer_JoZw · 2024-08-13
> > **Follow up**
> >
> > I would like to thank the authors for addressing my concerns and extending their computational evaluation.

---

### Author Rebuttal · Authors · 2024-08-06

We would like to thank the reviewers for their detailed feedback. We briefly summarize our responses here for clarity:

1. One general question was regarding the positioning of our work relative to prior work on pruning.
In this context, our work tries to provide the first rigorous analysis of approaches inspired by the Optimal Brain Surgeon paper of [1], achieving quadratic convergence rates in the case of an idealized algorithm for choosing masks. Surprisingly, we are also able to provide similar rates for a practical algorithm based on greedy TopK mask selection (Lemma 3).
In addition, our work directly connects to recent work on pruning: as discussed in Section 3.4, methods such as WoodFisher/WoodTaylor [2] and OBC [3] are special cases of our iterative framework. Generally, our iterative approach can be seen as complementary to methods investigating better one-shot Hessian approximations, such as CBS [4], as they can be directly plugged into our algorithm’s projection/selection steps.
To fully address this point, we have provided detailed comparisons with additional recent work suggested by the reviewers, such as CBS and CHITA[5] , in the individual responses, and will definitely expand this discussion and the citations in the next revision of our work.

2. Another general question regarded expanding the experimental comparisons. To address this, we provide additional experiments on Llama2-7B and Llama3-8B models, again providing significant improvements relative to the baselines (specifically, SparseGPT[6]). Moreover, we show that this improvement also holds on the MMLU LLM task, a subset of the suggested LLM-KICK benchmark. (We plan to further expand this comparison in the next revision.)

3. We provide the extra experiment results in the attached PDF file of this global rebuttal.

4. Beyond these two common points, we have provided answers to each individual reviewer's concerns.

We hope that our responses address the reviewers’ questions, and would be happy to continue the discussion during the rest of the rebuttal period.

[1] LeCun, Yann, John Denker, and Sara Solla. "Optimal brain damage." Advances in neural information processing systems 2 (1989).

[2] Singh, Sidak Pal, and Dan Alistarh. "Woodfisher: Efficient second-order approximation for neural network compression." Neurips 2020

[3] Frantar, Elias, and Dan Alistarh. "Optimal brain compression: A framework for accurate post-training quantization and pruning." Neurips 2022

[4] Yu, X., Serra, T., Ramalingam, S., & Zhe, S.. “The combinatorial brain surgeon: pruning weights that cancel one another in neural networks.”  ICML 2022

[5] Benbaki, Riade, et al. "Fast as chita: Neural network pruning with combinatorial optimization." ICML 2023

[6] Frantar, Elias, and Dan Alistarh. "Sparsegpt: Massive language models can be accurately pruned in one-shot."ICML 2023

---

### Decision · Program_Chairs · 2024-09-25

**Decision:**

Accept (poster)

**Comment:**

This paper proposes iterative approaches for pruning machine learning models, motivated by the iterative hard thresholding algorithm and the compressed sensing literature in conjunction with the Optimal Brain Surgeon framework. By employing second order information, the authors generalize previous approaches. Theoretical guarantees are derived for their pruning approach under suitable assumptions, and the numerical experiments support the authors' claims.

There is a consensus among reviewers that this work would be a good addition to the conference, and I agree.